# *Train, Mutate, or Reward?*
# A UNIFIED VIEW OF SUPERVISED ENSEMBLING FOR TIME SERIES ANOMALY DETECTION

## ABSTRACT

Time series anomaly detection (TSAD) is a long-standing and extensively studied problem with applications across a large panel of domains. Despite the maturity of the field, recent benchmark studies have revealed that no single detection method consistently outperforms others across diverse datasets. While model selection approaches (i.e., choosing the best detector for a given scenario) have shown promising results, their effectiveness remains inherently limited by the performance ceiling of existing individual detectors. To address this limitation, supervised ensembling offers a promising path to surpass individual detectors by learning to combine their strengths. In this work, we unify and formalize the problem of supervised ensemble-based anomaly detection in time series, and introduce three principled strategies for learning such ensembles: (1) classical Machine Learning, (2) Reinforcement Learning, and (3) Genetic Programming. We perform a rigorous comparative evaluation across these strategies using identical model components, inputs, and experimental conditions to ensure fairness. Our findings not only highlight the strengths and trade-offs of each approach, but also illuminate promising directions, paving the road for future research on this topic.

## 1 INTRODUCTION

Anomaly detection in time series data is a long-standing and critical task with broad applicability across various domains, including finance, industrial monitoring, healthcare, environmental science, and cybersecurity. Over the years, a wide range of methods have been developed, including statistical techniques (Li et al., 2020), unsupervised learning, and more recently, deep learning-based approaches (Chauhan & Vig, 2015; Kim et al., 2018). In particular, unsupervised anomaly detection (Goldstein & Dengel, 2012; Audibert et al., 2020) remains the dominant paradigm due to the scarcity of labeled anomalies in real-world datasets.

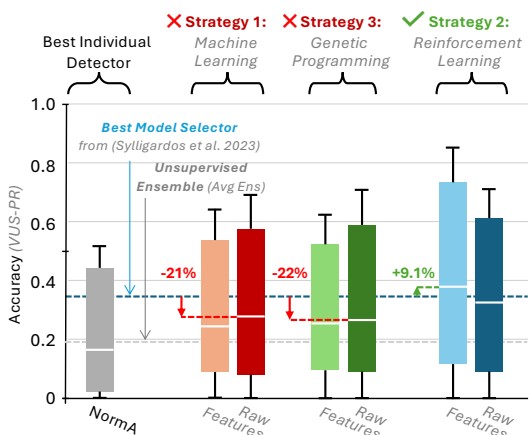

Figure 1: Comparison of the three strategies on the TSB-UAD benchmark (Paparrizos et al., 2022b).

However, recent benchmark studies (most notably the TimeEval (Schmidl et al., 2022), TSB-UAD (Paparrizos et al., 2022b), and TSB-AD (Liu & Paparrizos, 2025) benchmarks) have revealed a key insight: *no single detector consistently achieves top performance across diverse datasets and anomaly types*. This heterogeneity challenges the generalizability of individual methods and has motivated research into *model selection* approaches, where the best-performing detector is chosen based on time series characteristics. Although promising, such methods are intrinsically limited by the performance ceiling of the best available individual detector.

To overcome this ceiling, supervised ensemble learning offers an appealing alternative: rather than selecting a single detector, ensembles aim to combine the strengths of multiple detectors to achieve performance that exceeds any individual detector (Sylligardos et al., 2023). While ensemble techniques are widely studied in classification and regression, their application in the domain of time series anomaly detection, particularly under a supervised setting, remains under-explored and lacks a unified formalization and evaluation.We emphasize here that we do not assume that abundant labeled anomalies are available in general but rather investigate whether limited supervision can be leveraged to surpass the performance ceiling of unsupervised methods.

In this paper, we conduct a comprehensive study of **supervised ensembling strategies for time series anomaly detection**. We formalize the task as a supervised problem, where the objective is to learn a function that produces weights to combine individual detectors and improve detection performance. We explore this problem through three principled and diverse learning paradigms:

- **Classical Machine Learning (ML):** A given model trained on either raw time series sequences or pre-computed features (Lubba et al., 2019).
- **Reinforcement Learning (RL):** An agent (i.e., a model) that learns to combine detector scores over time by maximizing a reward given by the anomaly detection accuracy.
- **Genetic Programming (GP):** An approach where individuals (i.e., models) are evolved via mutation and selection, optimizing the anomaly detection accuracy as a fitness function.

We compare *feature-based* and *raw-based* input representations within each strategy to assess their impact on ensemble quality. All methods are evaluated on the TSB-UAD benchmark, using consistent test splits, detectors, and experimental conditions to ensure fairness and comparability. Crucially, our goal is not to compare architectures (CNN vs LSTMs) or different ensembling methods (Stacking (Wolpert, 1992) vs Weighted voting (Khan et al., 2024)) or hyperparameter sensitivity or even to change the detector pool but to benchmark our strategies upon a simple, reproducible baseline for supervised ensembling grounded in existing research.

To establish the value of supervised ensembling, we benchmark our strategies against: (i) the *best individual detector on TSB-UAD*, (ii) a naive *unsupervised average ensemble*, and (iii) the strongest *model selection* baselines from a recent experimental evaluation (Sylligardos et al., 2023).

Our findings, summarized in Figure 1, demonstrate that supervised ensembles can reliably outperform individual detectors and unsupervised ensembling, and in some cases, even surpass the best model selection methods (Sylligardos et al., 2023). These results highlight the potential of supervised learning in this context and open up promising directions for future work.

## 2 BACKGROUND AND RELATED WORK

In this section, we review time series anomaly detection literature and discuss recent automated solutions while detailing their limitations. First, a time series $T = [T_1, T_2, ..., T_L] \in \mathbb{R}^L$ is a sequence of real-valued numbers $T_i \in \mathbb{R}$, where $L = |T|$ is the length of $T$, and $T_i$ is the $i^{th}$ point of $T$. Local regions of the time series, known as subsequences $T_{i,\ell} \in \mathbb{R}^\ell$ of a time series $T$, is a subset of successive values of $T$ of length $\ell$ starting at position $i$, formally defined as $T_{i,\ell} = [T_i, T_{i+1}, ..., T_{i+\ell-1}]$. For a time series $T \in \mathbb{R}^L$, an anomaly detection method (or detector) $D$ returns an anomaly score sequence $S_T \in \mathbb{R}^L$.We note $N$ the number of detectors.

### 2.1 TIME SERIES ANOMALY DETECTION

Anomaly detection in time series is a crucial task for many relevant applications. Therefore, several methods (or detectors) have been proposed (Boniol et al., 2024). One type of anomaly detection method is *distance-based methods*, which analyze subsequences by utilizing distances to a given model to detect anomalies (Yeh et al., 2016; Breunig et al., 2000; Boniol et al., 2021a).

While methods in the previous category compute their anomaly score based on distances using raw time series elements (such as subsequences), *density-based methods* focus on detecting recurring or isolated behaviors by evaluating the density of points or subsequences within a specific representation space. This category can be divided into four sub-categories, namely *distribution-based*, *graph-based* (Boniol & Palpanas, 2020), *tree-based* (Liu et al., 2008), and *encoding-based*.

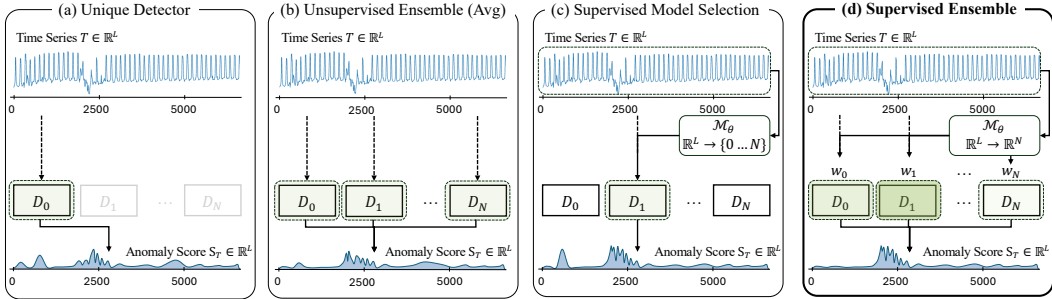

Figure 2: Time series anomaly detection: from unique detector (a) to supervised ensembling (d).

Finally, *prediction-based* approaches aim to detect anomalies by predicting the expected normal behaviors based on a training set of time series or sub-sequences (containing anomalies or not). Methods in this category detect anomalies using prediction errors. More specifically, such category can be divided into *forecasting-based methods* (Malhotra et al., 2015), and *reconstruction-based methods* (Sakurada & Yairi, 2014). A more detailed review is in the Appendix A.1.

## 2.2 ONE UNIQUE DETECTOR IS HOPELESS

Recently, several experimental evaluations for anomaly detection in time series have been proposed (Schmidl et al., 2022; Paparrizos et al., 2022b). Such benchmarks provide a large collection of time series from various domains and evaluate multiple methods spanning all the categories mentioned above. However, these experimental evaluations led to the same conclusion: no single method is universally best across all domains. This is explained by the following two reasons:

**Heterogeneity in anomaly types:** Time series anomalies are either *point*, *contextual*, or *collective*. Such heterogeneity in anomaly types makes the anomaly detection task challenging. Moreover, even for time series with the same anomaly types, we observe that the most accurate models are all different (Sylligardos et al., 2023).

**Heterogeneity in time series structures:** On top of heterogeneity in anomaly types, we need to differentiate time series containing *single* or *multiple* anomalies (with different or similar anomalies). For instance, methods based on neighbor distance computation, such as LOF, are very accurate in detecting *single* or *multiple different* anomalies, but less accurate for *multiple similar*.

## 2.3 TOWARDS AUTOMATED SOLUTIONS

A solution to the limitations mentioned above is to apply **model selection** (depicted in Figure 2(c)) based on time series characteristics (Sylligardos et al., 2023). The goal is to train a model to automatically select the best detectors for a given time series. However, model selection methods are inherently constrained by the detectors they select from. Since these approaches operate by choosing an existing detector, their performance is fundamentally bounded by the best detector within the candidate pool. As such, model selection cannot produce results that exceed the capabilities of its individual components (in this paper, we refer to the *Oracle* as the best theoretical model selector). This limitation arises when no single detector performs well across all datasets or anomaly types.

A solution to this limitation is to apply an **unsupervised ensemble** to the anomaly scores produced by all the detectors (Figure 2(b)). While several unsupervised ensembling techniques have been proposed (Aggarwal & Sathe, 2015), the *Averaging* strategy is the most robust and low-risk strategy (Aggarwal & Sathe, 2015). However, despite offering simplicity (Goswami et al., 2022; Schmidl et al., 2024) and robustness, this approach remains fundamentally agnostic to the time series and the contextual reliability of each detector (i.e., all treated equally). As a result, they are unable to emphasize the strengths of high-performing detectors or mitigate the weaknesses of less reliable ones. In contrast, a *supervised ensemble* (illustrated in Figure 2(d)) can learn to combine detector outputs based on actual anomaly patterns, potentially surpassing the performances of individual detectors, naive ensembles, and model selection.

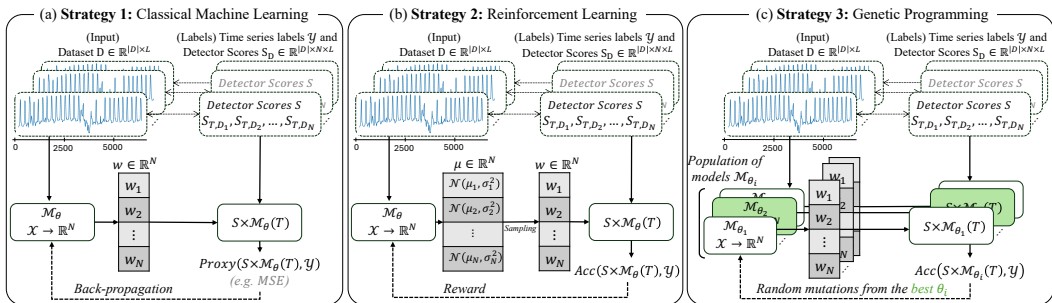

Figure 3: Overview of the different strategies considered for supervised ensembling.

## 2.4 SUPERVISED ENSEMBLING: *Problem Definition*

For a time series of length $L$, the supervised ensemble learning paradigm is defined as follows: the input data $X$ is either a feature vector in $\mathbb{R}^F$, where $F$ is the number of features, or a raw time series in $\mathbb{R}^L$. Given $X$, the weights $W$ applied to each individual detector are sampled from a trained model $\mathcal{M}_{\theta_*} : X \to p_X \in \mathcal{P}(\mathbb{R}^N)$ which outputs a probability distribution over the set of weighting coefficients for the $N$ detectors. The anomaly scores from the detectors are denoted by $S \in \mathbb{R}^{L \times N}$, and the corresponding ground truth label is represented by $Y \in \{0, 1\}^L$. Finally, we note $Acc$, the anomaly detection metric (such as VUS-PR). The optimal model parameters, defined by $\theta_* \in \Theta := \mathbb{R}^{d_\mathcal{M}}$ where $d_\mathcal{M}$ is the number of parameters, are obtained as follows:

$$\theta_* = \arg\max_{\theta \in \Theta} \mathbb{E}_{(X,S,Y),\, W \sim \mathcal{M}_\theta(X)} \left[ Acc\big(S \times \widetilde{W}, Y\big) \right]. \tag{1}$$

Since $\theta_*$ is unknown, we aim at maximizing the expected accuracy metric $Acc$ through appropriate learning strategies. The term $\mathcal{S} \times \widetilde{W} \in \mathbb{R}^L$ represents the output of the supervised ensemble model, i.e., the weighted anomaly scores, in which $\widetilde{W}$ is the weights vector $W$ normalized by its norm. Therefore, the goal is to identify the most effective strategy for optimizing $\mathcal{M}_\theta$ as defined in Equation 1. The most effective strategy can be **Deterministic** (such as ML or GP) or **Stochastic** (such as RL). We provide a reformulated version of Equation 1 for the deterministic case in Appendix A.3.

## 3 TSAD SUPERVISED ENSEMBLING: *Three Strong Baselines*

In this section, we present three strong baselines for supervised ensembling for time series anomaly detection. For fair comparison, we introduce a generic pipeline as a foundation for all strategies.

### 3.1 A GENERIC PIPELINE

To consistently evaluate the three supervised ensembling strategies, we define a generic training and evaluation pipeline. Each time series in the dataset is either (i) converted into a set of $F$ statistical features, resulting in one feature vector per time series, or (ii) reduced to the first window of fixed length $\ell$ if using the raw time series data directly. The model (as defined by the chosen strategy) outputs a weight vector $W \in \mathbb{R}^N$, where $N$ is the number of detectors. We compute the final ensemble anomaly score for a given time series $T$ as a weighted average of the detector scores using the predicted weight normalised noted $\widetilde{W}$. Contrary to Mixture of Experts (MoE) (Yuksel et al., 2012) in our framework, the experts are a diverse set of pre-existing, heterogeneous, and pre-trained algorithms. The pipeline is summarized in Algorithm 1 in the Appendix A.4.

### 3.2 STRATEGY 1: *Classical Machine Learning*

We first consider the most straightforward path to solve the optimization problem: a gradient method minimizing an objective defined as closely as possible to the anomaly detection accuracy (illustrated in Figure 3(a)). The non-differentiability of the time series anomaly detection accuracy measure AUC-PR or VUS-PR (Paparrizos et al., 2022a) imposes on us to choose a differentiable proxy, such

as the Mean Squared Error (MSE). However, selecting a differentiable proxy introduces inherent limitations, as we are no longer optimizing the true objective (i.e., the anomaly detection accuracy). Instead, we rely on a surrogate that may not faithfully reflect the nuances of the actual target. Formally, the problem initially formulated in Equation 1 is the following:

$$\theta_* = \arg\max_{\theta \in \Theta} \mathbb{E}_{(X,S,Y)}\Big[\text{Proxy}\big(S \times \widetilde{M_\theta}(X), Y\big)\Big]. \tag{2}$$

Where $\widetilde{M_\theta}(X)$ are the normalized output (weights) of $M_\theta$. In practice, we consider the MSE as the Proxy. It is also important to highlight that the pipeline used in Strategy 1 closely resembles the model selection approach proposed in Sylligardos et al. (2023). The key difference lies in the fact that we are performing regression (i.e., predicting continuous weights for each detector) rather than classification. While framing the problem as a classification task may be more natural (considering the labels we are generating), adopting a regression-based approach ensures consistency with Strategies 2 and 3. This alignment allows for a fairer comparison between the strategies, focusing solely on the learning paradigm employed.

### 3.3 STRATEGY 2: *Reinforcement Learning*

In order to directly optimize a non-differentiable evaluation metric (such as VUS-PR) without the need to handcraft a surrogate loss, we consider a reinforcement learning (RL) strategy (illustrated in Figure 3(b)). The RL paradigm offers the flexibility to treat the evaluation metric as a reward, using it to guide the learning process via policy gradient methods (Sutton et al., 1998). In such a framework, the agent is designed to (i) *observe a state* (i.e., each time series or its feature representation), (ii) *take an action* (predicting the set of weights used to aggregate the anomaly scores of the detectors) and then (iii) *collect the associated reward* (i.e., the anomaly detection accuracy).

The objective is to learn a policy $\pi_\theta$ that guides the actions of the agent according to the state: it associates to any state X (the input of a given model, such as features or raw subsequences) the probability distribution of the action W (the weights) such that the reward (the anomaly detection accuracy) is maximum. The policy could be written $\mathcal{M}_\theta$ for coherence with Equation 1 but we choose the following RL notation: $\pi_\theta : X \mapsto \{\mathcal{N}(W_i; \mu_\theta^i, (\sigma_\theta^i)^2)\}_{1 \leq i \leq N}$ where $\mu_\theta^i$ is the mean and $\sigma_\theta^i$ the standard deviation of the normal distribution. The policy can be written with respect to the model $M_\theta$ as $\pi_\theta : X \mapsto \{\mathcal{N}(W_i; M_\theta(X)_i, (\sigma_\theta^i)^2)\}_{1 \leq i \leq N}$ where $M_\theta(X)_i$ and $W_i$ are respectively the $i^{th}$ output of the model and component of the weight vector. The reward is defined by $R(X, W) := Acc\big(S \times \widetilde{W}, Y\big)$ with the notations of equation 1. We note here that there is no spatial dependency between the states; each state-action-reward tuple is treated independently.

In practice, we use the Proximal Policy Optimization (PPO) algorithm (Schulman et al., 2017). In such an algorithm, a shared critic network is used to estimate the value function $V_{\pi_\theta}(X) = \mathbb{E}_{W \sim \pi_\theta(X)}[R(X, W)]$, which is then used to compute the advantage function $A_{\pi_\theta} = R(X, W) - V_{\pi_\theta}(X)$ emphasizing how good (in terms of reward) was the action $W$ compared to the average reward of the action given by the current policy $\pi_\theta$. The objective function is defined as follows:

$$L^{\text{CLIP}}(\theta) = \mathbb{E}_{\pi_\theta}\big[\min\big(r(\theta)A_{\pi_\theta}, \text{clip}(r(\theta), 1 - \epsilon, 1 + \epsilon)A_{\pi_\theta}\big)\big] \tag{3}$$

where $\epsilon > 0$, and $r(\theta) := \pi_\theta(X)(W)/\pi_{\theta_{\text{old}}}(X)(W)$ relates the evolution of the likelihood of the action $W$ at the state $X$. clip in Schulman et al. (2017) bounds the value $r(\theta) \in [1 - \epsilon, 1 + \epsilon]$.

### 3.4 STRATEGY 3: *Genetic Programming*

In contrast to Strategies 1 and 2 that learn via gradient descent, genetic programming offers a population-based, gradient-free optimization paradigm (illustrated in Figure 3(c)). Like reinforcement learning, it does not require the objective function to be differentiable. However, genetic algorithms do not rely on policy optimization or estimating gradients. Instead, it directly explores the solution space through an evolutionary process (Mitchell, 1998). Thus, it complements the other strategies by offering an almost model-free baseline. This complementarity is further supported by the literature on hybrid frameworks, such as Evolutionary Algorithm Reinforcement Learning (EARL) (Moriarty et al., 1999).

The goal is to evolve a population of candidate solutions, where each individual represents a model $\mathcal{M}_\theta$ (i.e, a vector of parameters $\theta$ for different models with the same architecture) used to predict

Table 1: Experimental setup for the evaluation of Ensemble Strategies

| | Strategy 1 (ML) | Strategy 2 (RL) | Strategy 3 (GP) |
|---|---|---|---|
| **Datasets and Detectors** **Train/Test Split** | TSB-UAD (Paparrizos et al., 2022b) (16 datasets, 12 detectors) *In-distribution* : 70% Train, 30% Test ; *Out-of-distribution* : Leave-One-Dataset-Out | | |
| **Input Representations** | Raw windows of length 128 *or catch22* features (Lubba et al., 2019) | | |
| **Core Model** *(Features)* **Core Model** *(Raw)* | Best features-based model in (Sylligardos et al., 2023) :    Multi-Layer Perceptron (MLP) Best window-based model in (Sylligardos et al., 2023) :  Convolutional Neural Network (CNN) | | |
| **Optimization Strategy** **Loss / Objective** **Optimization Details** | Backpropagation MSE ADAM Optimizer | PPO (Schulman et al., 2017) AUC-PR or VUS-PR as reward PPO with clipped surrogate loss | Genetic Algorithm AUC-PR or VUS-PR as fitness `pygad.GA` (Gad, 2023) |

our targeted weights. The quality of each individual is evaluated using the anomaly detection metric $Acc$ (such as AUC-PR or VUS-PR), guiding the evolutionary search toward weight configurations that produce better ensemble performances for our anomaly detection task. More formally, $P^k = \{\theta_1^k, \theta_2^k, \ldots, \theta_N^k\}$ is the *population* at generation $k$, composed of $N$ candidate solutions. Each $\theta_i^k \in \mathbb{R}^d$ is called an *individual* and is an approximation of $\theta_*$ in equation 4.

In our case, each $\theta_i^k$ corresponds to a flattened version of the weights of a neural network. Thus, genetic algorithms explore the space of network parameters without relying on gradient information. Instead, it keeps the fittest individuals to generate a new population. The fitness function $f : \mathbb{R}^d \to \mathbb{R}$ of each model is evaluated based on a given accuracy measure $Acc$ and is defined as follows:

$$f(\theta) := \mathbb{E}_{(X,S,Y)}\big[Acc\big(S \times \widetilde{M}_\theta(X), Y\big)\big] \quad ; \quad \theta_* = \arg\max_{\theta \in \Theta} f(\theta) \tag{4}$$

The selection phase chooses the best individuals (based on fitness) to serve as parents. Among several strategies, we use the steady-state approach, in which only the least fit individuals are replaced each generation, while the best are retained. Formally, let $M \leq N$ and define $\{\theta_1^k, \ldots, \theta_M^k\} \subset P^k$ as the $M$ best individuals at generation $k$. These parents produce offspring for the next generation through crossover (each child inherits half of its parameters from each parent) and mutation (10% of a child's parameters are perturbed by adding a random value between $[-1, 1]$).

## 4 EXPERIMENTAL EVALUATION

We now evaluate the three strategies described above by answering the following questions:

**(Q1) How do supervised ensembles compare to traditional baselines?** We benchmark our strategies against individual detectors, unsupervised ensembles, and model selection.

**(Q2) Do supervised ensembles really ensemble?** We assess whether the supervised ensemble mimics model selection methods (selecting only one detector) or combines meaningful detectors (leading to higher performances than the best detector on each time series).

**(Q3) How do the strategies scale?** We measure training and detection time for all strategies.

### 4.1 EXPERIMENTAL SETUP

Our experimental setup for evaluating the three proposed ensemble strategies is summarized in Table 1. Overall, strategy 1 relies on simple backpropagation using the ADAM optimizer with mean squared error (MSE) loss. Strategy 2 employs the Proximal Policy Optimization (PPO) algorithm (Schulman et al., 2017), where the core model acts as the policy network. Strategy 3 uses genetic programming via the `pygad.GA` library (Gad, 2023) to evolve model weights directly.

Additionally we benchmark their performance against four baselines: (i) 12 individual detectors provided in the TSB-UAD benchmark (Paparrizos et al., 2022b), (ii) an unsupervised ensemble baseline computed as the average of all detectors' outputs (called **Avg Ens** in this paper), and (iii) the best model selection method identified in (Sylligardos et al., 2023) (called **Best MS** in the paper) (iv) the best theoretical model selector, i.e., selecting the best detector for each time series (called **Oracle** in the paper). These comparisons enable a comprehensive assessment of the benefits and

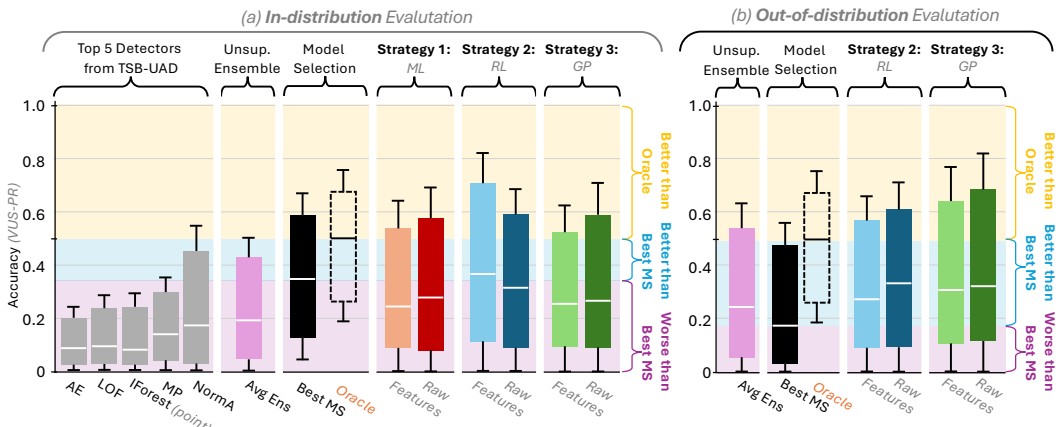

Figure 4: Accuracy comparison (for (a) *in-distribution* averaged over 5 random seeds and (b) *out-of-distribution*) of Strategy 1, 2, and 3 versus individual detectors, unsupervised ensemble, and the best model selector. The box extends from the first quartile (Q1) to the third quartile (Q3) with the median line. Whiskers cover the data range up to $0.24 \times$ (Q3-Q1) from the box edges.

limitations of our proposed strategies relative to established methods. We provide more technical details in our repository [1] for reproducibility purposes.

## 4.2 HOW DO SUPERVISED ENSEMBLES COMPARE TO BASELINES?

In this section, we first compare the different strategies against the baselines (*Individual detectors*, *Avg Ens*, *Best MS*, and *Oracle*) and we identify which strategy is the most promising. In Figure 4, all strategies produce weights in $[0, 1]$. The scenario where ML and GP infer detector weights in $[-1, 1]$ is discussed in Appendix. To achieve the comparison between supervised ensembling and the baselines we conduct two experiments: in the first (Figure 4(a)), we evaluate the anomaly detection accuracy (VUS-PR) of all strategies over the entire TSB-UAD benchmark (i.e., *in-distribution* setting). In such a setting, both the training and the test sets contain time series from all 16 datasets of TSB-UAD. In the second experiment (Figure 4(b)), we evaluate the performance of the models in an *out-of-distribution* setting (i.e., evaluated on a dataset not used in the training set) serving as a proxy for robustness assessment in label-scarce scenarios and demonstrating that supervision can be transferable and practically valuable. In this scenario, we leave one entire dataset out for the test and use the 15 remaining datasets for training. Based on the performance in the first experiment, we exclude Strategy 1 from the second experiment.

Table 2: Variability study of the mean performance of each strategy across 5 random seeds in the *in-distribution* settings (mean ± std). Strategy 2 (RL) on features consistently outperforms the other approaches.

| Method | VUS-PR |
|---|---|
| **Strategy 1 (ML)** | |
| Features | $0.336 \pm 0.019$ |
| Raw | $0.347 \pm 0.005$ |
| **Strategy 2 (RL)** | |
| Features | $\mathbf{0.440 \pm 0.004}$ |
| Raw | $0.382 \pm 0.008$ |
| **Strategy 3 (GP)** | |
| Features | $0.332 \pm 0.020$ |
| Raw | $0.356 \pm 0.010$ |
| Avg. Ens. | 0.2734 |
| Model Selection (MS) | 0.3819 |
| **Oracle** | 0.4885 |

We first observe in Figure 4(a) that, for *in-distribution*, all supervised ensembling strategies are significantly outperforming individual detectors and *Avg Ens*. Moreover, for *in-distribution*, Strategy 2 (RL) on features outperforms *Best MS*. In particular, Strategy 2 on features outperforms the median of the best model selector, and the distribution of accuracy performances reaches higher values than the *Oracle*. Although lower in terms of median, the latter underlines that the model selection *barrier*

---
[1]Our repository: link

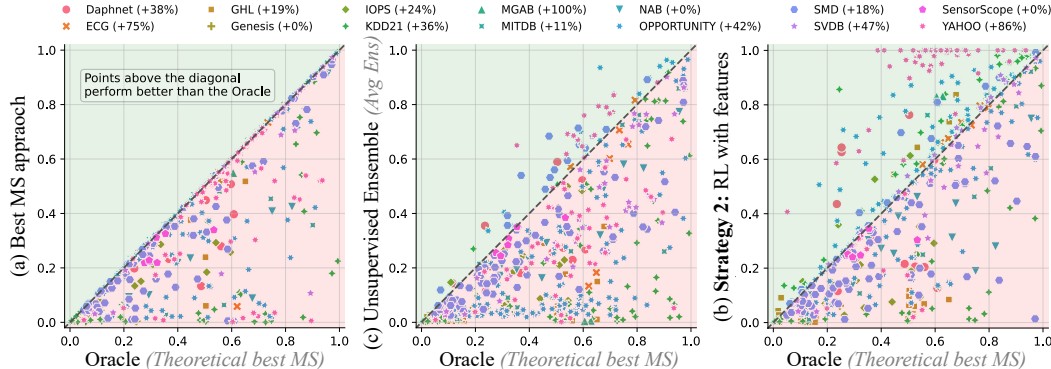

Figure 5: (a) *Best MS*, (b) *Avg Ens*, (c) Strategy 2 (with features) against the *Oracle*. % of time series per datasets on which RL (with features) is outperforming the *Oracle*.

can be surpassed by some supervised ensembling strategies in specific datasets (more details in Section 4.3). In addition, Figure 4(a) demonstrates that, for *in-distribution*, the choice of the strategy (choosing RL versus GP or ML) appears more relevant than the type of input (*feature* versus *raw*) as the gap between supervised ensemble strategy performances is bigger than the differences between feature and raw-subsequences for a given strategy. The strong performances of Strategy 2 (RL) are likely due to its guided optimization scheme on the true TSAD metric VUS-PR. These observations are reinforced by the variability results displayed in Table 2 across 5 different seeds, assessing the stability of the mean accuracy of the methods, and thus the performance hierarchy.

Finally, Figure 4(b) leverages multiple interesting observations. As already observed in Sylligardos et al. (2023), ensembling techniques (*Avg Ens*) are outperforming model selection approaches in the *out-of-distribution* setting. This observation is confirmed in Figure 4(b) as the supervised ensemble strategies (RL and GP) outperform both the average ensemble and the *Best MS*. Surprisingly, unlike the *in-distribution* case, the *out-of-distribution* scenario favors the choice of the data type over the strategies. Indeed, whether we consider strategy 2 or 3, the accuracies are globally similar; however, when applying the strategies on raw data, the performances are significantly higher than those of feature-based methods. The latter can be explained by the following two reasons: (i) In *out-of-distribution* scenario, ensembling more detectors (such as the *Avg Ens* that ensembles all detectors equally) is safer than selecting a few (such as the *Best MS* that picks only one). We observe in practice that supervised ensembling strategies applied to features tend to select (i.e., with non-zero weights) fewer detectors than the same strategies applied to raw subsequences. (ii) The feature space is sparser than the raw subsequences, which might lead to lower generalisability capabilities.

> **Answer to Q1:** *In the in-distribution case, supervised ensemble strategies outperform individual detectors and the unsupervised ensemble. Strategy 2 (RL) on features is the only supervised ensembling pipeline that outperforms the best model selection methods in Sylligardos et al. (2023). However, in the out-of-distribution case, Strategies 2 and 3 (RL and GP) are outperforming both the unsupervised ensemble and the best model selection method. Finally, the choice of strategy prevails over the choice of data type in the in-distribution case, and the choice of data type has priority over the strategy in the out-of-distribution case.*

## 4.3 DO SUPERVISED ENSEMBLES REALLY ENSEMBLE?

In the previous section, we observed that a supervised ensemble (Strategy 2 on features) can achieve higher performance than the *Oracle*. Therefore, we conduct a per-time series comparison in this section between Strategy 2 and the *Oracle*. Thus, we gather the performances of *Best MS*, *Oracle*, *Avg Ens*, and the best supervised ensemble method (*RL with features*). Figure 5 depicts a pairwise comparison between them (i.e., each point in these figures is a time series).

First, Figure 5(a) shows that model selection has a glass ceiling imposed by the performances of each individual detector: the *best MS* can not outperform the *Oracle*, by definition. Moreover, the average

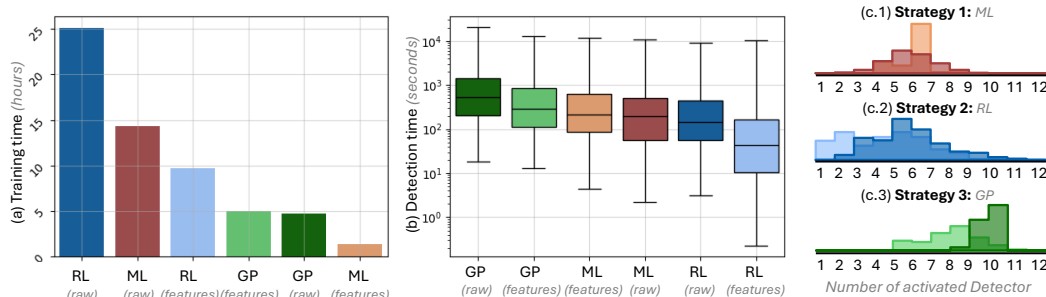

Figure 6: (a) Training time, (b) Detection time (i.e., both computing the weights and running the activated detectors), and (c) Number of activated detectors per strategy.

ensemble (in Figure 5(b)) fails to convincingly overcome this observation: even though a small set of time series are above the diagonal (i.e., *Avg Ens* outperform the *Oracle*), we observe no flagrant progress and note that, unlike *Best MS*, many time series (e.g., from the OPPORTUNITY dataset) suffer from poor anomaly detection performances (VUS-PR$\leq 0.1$) where the *Oracle* stays efficient (VUS-PR$\geq 0.4$). Contrary to the unsupervised average ensembling, the supervised RL ensembling approach on features (in Figure 5(c)) demonstrates promising results. For instance, the *RL on feature* strategy outperforms the *Oracle* for $42\%$ of the OPPORTUNITY time series. Furthermore, this is also evident in the YAHOO dataset (containing time series with point anomalies only), where an important fraction ($86\%$) of the time series shows that the *RL on features* outperforms the *Oracle*. Finally, we also observe from Figure 5(c) that the supervised ensemble is not accurate for time series with a very highly inaccurate individual detector (i.e., not points in the top-left corner of Figure 5(c)). In particular, as soon as the *Oracle* is performing poorly, *RL on features* fails to beat the *Oracle*. However, we emphasize that breaking the model selection *barrier* (i.e., being above the diagonal) becomes more frequent as the *Oracle* performs better.

> **Answer to Q2:** *Supervised ensembling (Strategy 2 on feature) embodied the expectations associated with ensembling methods: it managed to break the model selection barrier by outperforming the Oracle (i.e., more accurate for at least $50\%$ of the time series) on the MGAB ($100\%$), YAHOO ($86\%$) and ECG ($75\%$) datasets, and being challenging (i.e., more accurate for at least $40\%$ of the time series) on SVDB ($47\%$) and OPPORTUNITY ($42\%$).*

### 4.4 HOW DO THE DIFFERENT STRATEGIES SCALE?

We now analyze the training and detection time of all three supervised ensembling strategies. The training times (depicted in Figure 6(a)) relate to the time needed to train and test the models entirely. The detection time (depicted in Figure 6(b)) depicts the time required to compute the weights and run the activated detectors (i.e., detectors associated with non-zero weights).

Figure 6(a) shows that methods handling raw subsequences are slower than methods handling features for Strategies 1 and 2 (ML and RL) and equivalently slow on Strategy 3 (GP). The latter is explained by the following two reasons: (i) the time required to compute the weights and (ii) the experimental setup. Indeed, inferring the weights on raw subsequences is slower, as the CNN model shared across all three strategies has $\sim 100$ times more parameters than the MLP used for methods that handle features. However, the experimental setup compensates for this by limiting the number of evaluations (more details in Appendix A.5).

On the contrary, Figure 6(b) shows that the detection time is not directly affected by the input type. As observed jointly in Figure 6(b) and (c), The Detection time is inversely proportional to the number of activated detectors. Thus, the sparsity of the strategies on the detector weights (i.e., producing few non-zero weights) is key to significantly reducing detection time (in Figure 6(c), distribution shifted to the left means sparser weights prediction). More precisely, as illustrated in Figure 6(c), RL and GP strategies activate fewer detectors when considering features as input and thus minimize the detection time. We also observe that gradient-based methods like ML and RL (with PPO) induce

sparser model activations. All these remarks highlight a key result: *RL on features* is not only the best supervised ensemble technique in the *in-distribution* case but also the fastest strategy.

The sparsity of the activated detector is also strongly correlated to the performance in the *out-of-distribution* setting. In fact, we previously noticed on Figure 4(b) that model selection (the sparsest supervised approach) showed very weak performances. Conversely, we emphasized how supervised ensembling approaches that consider raw data were more effective than those relying on features. The latter can be explained by the low sparsity of raw-based strategies (2 and 3 mainly) that mitigates the risk of relying on a specific detector that might fail on the new data.

> **Answer to Q3:** *Supervised ensembling produces very different activated detector distributions, which have a significant impact on detection time and scalability (the sparser, the faster), as well as on the out-of-distribution scenario results (the denser, the better).*

## 5 CONCLUSION AND KEY TAKEAWAYS

We conduct a comprehensive study of supervised ensembling strategies for time series anomaly detection. We explore this problem through three principled and diverse learning paradigms (ML, RL, and GP) and evaluate these three strategies in terms of anomaly detection accuracy and execution time. Overall, our key takeaways are as follows:

**(T1)** supervised ensembling systematically outperforms model selection in the *out-of-distribution* setting and manages to do it in the *in-distribution* setting with Strategy 2 (RL) with features as input.

**(T2)** strategy 2 (RL) with features as input, successes in breaking the model selection barrier on several datasets.

**(T3)** the sparser the supervised ensembling methods are, the faster. However, sparser outputs imply poor *out-of-distribution* performances.

These results highlight the potential of supervised learning in this context and open up promising directions for future work, particularly in the context of Reinforcement learning, as a strong supervised ensembling baseline for time series anomaly detection.

## 6 REPRODUCIBILITY STATEMENT

For the sake of reproducibility, we provide a link to our anonymous source code for the experiments, allowing it to be analysed and reproduced in detail. In addition to that, we specify all the key parameters for every strategy in Section 4.1 and in the Appendix A.5.

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

# A  APPENDIX

This sections contains additional resources on the literature review (Section A.1 and Section A.2), the problem formulation (Section A.3), the generic pipeline considered in our experimental evaluation (Section A.4), the experimental setup (Section A.5), and additional experimental analysis (from Section A.6).

## A.1  TIME SERIES ANOMALY DETECTION LITERATURE REVIEW

Anomaly detection in time series is a crucial task for many relevant applications. Therefore, several methods have been proposed in the literature (Boniol et al., 2024). One type of anomaly detection method is *distance-based methods*, which analyse subsequences by utilizing distances to a given model to detect anomalies. In this category, we can identify three sub-categories. The first is *discord-based*. These methods focus on analysing subsequences for the purpose of detecting anomalies in time series, primarily by utilizing nearest neighbour distances among subsequences (Yeh et al., 2016). The second sub-category is *proximity-based*. These methods focus on estimating the density of specific types of subsequences to either extract normal behaviour or isolate anomalies. Since a subsequence can be seen as a multidimensional point (with the number of dimensions corresponding to the subsequence length), general outlier detection methods can be applied for time series anomaly detection (Breunig et al., 2000). The last category is *clustering-based*, which comprises methods that utilize the distance to a given clustering partition to detect anomalies. In this sub-category, NormA, which first clusters data to obtain the normal behaviour (Boniol et al., 2021a;b), has demonstrated strong performance.

While the previously mentioned methods compute their anomaly score based on distances using raw time series elements (such as subsequences), *density-based methods* focus on detecting recurring or isolated behaviours by evaluating the density of points or subsequences in a specific representation space. This category can be divided into four sub-categories, namely *distribution-based*, *graph-based*, *tree-based*, and *encoding-based*. Among them, Isolation Forest (Liu et al., 2008), a *tree-based* method grouping points or subsequences into different trees, and Series2Graph, a *graph-based* method that converts the time series into a graph to facilitate the detection of anomalies (Boniol & Palpanas, 2020), have been shown to work particularly well for the time series anomaly detection task (Boniol & Palpanas, 2020).

Furthermore, *forecasting-based methods*, such as recurrent neural network-based (Malhotra et al., 2015) or convolutional network-based (Munir et al., 2019), have been proposed for this task. These methods use the past values as input, predict the following one, and use the forecasting error as an anomaly score. Such methods are usually trained on time series without anomalies, or make the assumption that the anomalies are significantly less frequent than normal behaviours.

Finally, *reconstruction-based methods*, such as auto-encoder approaches (Sakurada & Yairi, 2014), are trained to reconstruct the time series and use the reconstruction error as an anomaly score. As both forecasting and reconstruction-based categories detect anomalies using prediction errors (either forecasting or reconstruction error), we can group them into *prediction-based methods*.

## A.2  ANOMALY TYPES

*Point* anomalies refer to data points that deviate remarkably from the rest of the data. Similarly, *contextual* anomalies refer to data points within the distribution's expected range (in contrast to point anomalies) but that deviate from the distribution's expected behavior given a specific context (e.g., a window). *Collective* anomalies refer to sequences of points that do not repeat a typical (previously observed) pattern. The first two categories, namely, point and contextual anomalies, are referred to as *point-based* anomalies, whereas *collective* anomalies are referred to as *subsequence* anomalies.

## A.3  DETERMINISTIC VERSUS STOCHASTIC LEARNING STRATEGIES

**Deterministic models: ML and GP approaches** The deterministic case appears when a function embodies the model, we will note $M_\theta : X \mapsto M_\theta(X) \in \mathbb{R}^N$. In practice, $M_\theta$ is implemented as a network, either an MLP or a CNN, depending on whether the data are features or a raw time-series

window. More formally, the model generating the weights can be defined with $\mathcal{M}_\theta(X) := \delta_{M_\theta(X)}$ where $\delta_{M_\theta(X)}$ is the Dirac distribution at the network output $M_\theta(X)$. The optimization problem can thus be reformulated with

$$\theta_* = \arg\max_{\theta \in \Theta} \mathbb{E}_{(X,S,Y)}\left[Acc\big(S \times \widetilde{M_\theta}(X), Y\big)\right]. \tag{5}$$

To solve this optimization task, we want to use gradient methods because they are simple to implement and effective in most cases. However, $Acc$ (AUC-PR or VUS-PR) is not differentiable, and facing this issue, we consider two approaches: Machine Learning and Genetic Algorithms. The former chooses to change the metric to a differentiable proxy, such as the *Mean Squared Error* (MSE). The latter instead decides to keep the metric of interest and forget about gradient methods for keeping the best random mutation of $\theta$ over generations to target $\theta_*$.

**Stochastic models: RL approach** The main problem described by equation 2.4 is exactly the stochastic case (the deterministic one is simply a particular case). More specifically, the RL strategy outputs a normal distribution specific to each individual detector, from which the weights are later sampled. We will see in the section dedicated to RL how we can, surprisingly, take the gradient of the main objective and how RL introduces not only a stochastic approach, but an entire paradigm.

### A.4 GENERIC PIPELINE

This section contains additional resources on the generic pipeline considered in our experimental evaluation. We recall that the weights $W$ are noted $\widetilde{W}$ when normalized.

---

**Algorithm 1** Generic Pipeline for Supervised Ensembling

---

**Require:** Dataset of time series $\mathcal{D}$, window size $\ell$, number of detectors $N$, model $\mathcal{M}_\theta$, training strategy
**Ensure:** Trained model parameters $\theta_*$, anomaly scores for test time series
 1: Split $\mathcal{D}$ into $\mathcal{D}_{\text{train}}, \mathcal{D}_{\text{val}}, \mathcal{D}_{\text{test}}$
 2: **for** each time series $T$ in $\mathcal{D}_{\text{train}} \cup \mathcal{D}_{\text{val}}$ **do**
 3:     **if** using raw time series **then**
 4:         collect the first window of size $\ell$ of $T$
 5:     **else if** using features **then**
 6:         Extract $F$ features from $T$ as input vector
 7:     **end if**
 8: **end for**
 9: Train model $\mathcal{M}_\theta$ on $\mathcal{D}_{\text{train}}$ and early stop on $\mathcal{D}_{\text{val}}$
10: Obtain final parameters $\theta_*$
11: **for** each test time series $T$ in $\mathcal{D}_{\text{test}}$ **do**
12:     **if** using raw time series **then**
13:         Select the first window of size $\ell$ of $T$ and predict the weight vector $W$
14:     **else if** using features **then**
15:         Extract features and predict the weight vector $W$
16:     **end if**
17:     Retrieve detector scores matrix $\mathcal{S} \in \mathbb{R}^{L \times N}$
18:     Compute ensemble score: $\hat{\mathcal{S}} = \mathcal{S} \times \widetilde{W}$
19: **end for**

---

### A.5 EXPERIMENTAL SETUP

**Technical setup.** All experiments were conducted on a server with Cascade Lake Intel Xeon 5217 8 cores, 3-3.7GHz CPU, and Nvidia V100 32GB GPU.
**Global Parameter settings** We use the same 70/30 split of the TSB-UAD benchmark as in Sylligardos et al. (2023) for comparison fairness. All methods were early-stopped if there was no evaluation improvement for half the total evaluation number.
**ML parameter settings**. On the features, the MLP was trained with a learning rate of $10^{-2}$ and a batch size of 32. As for the raw data, the CNN is made of 2 blocks, a GAP layer, and its kernel size

is 3. It was trained with a learning rate of $10^{-4}$ and a batch size of $64$. Both models were trained for 50 epochs.

**RL parameter settings**. Considering the parameters naming of Stable Baseline3, for both features and raw data, we fix the learning rate to $3.10^{-3}$, run the methods with 720000 total time steps (total amount of states visited, windows/features in our formulation), 20 epochs every 1024 steps, and a batch size of 128. The experiments on the raw data are evaluated 10 times along the way, and this number is raised to 100 for the features (since they were less time-consuming). Weights $W$ are clipped to 0-1. This decision is motivated empirically to favour convergence stability. We provide additional experiments with unclipped weights.

**GP parameter settings**. Both MLP and CNN were trained for 20 generations, 20 solutions, 5 parents mating, and a random sample of 256 time series to evaluate the fitness of each individual (for time efficiency purposes).

**Evaluation metrics.** For anomaly detection accuracy, we consider the measure VUS-PR (Paparrizos et al., 2022a). As detailed in Boniol et al. (2025), this metric is specifically designed to account for imprecise, noisy, misaligned, or time-delayed labels, ensuring that our reported performance is robust to the inherent ambiguity of anomaly timestamps. Concerning execution time evaluation, we mean by "execution time" the time required to perform all the training/testing phases (*i.e.*, the entire training pipeline). Finally, we will look at the inference time of each strategy, *i.e*, the time needed for a model (MLP or CNN) to output a vector for a given input and a given paradigm (ML, RL, GP).

**Features extraction and preprocessing.** Feature extraction is performed using the official pycatch22 library Lubba et al. (2019), ensuring a standardized, reproducible, and computationally efficient implementation. While feature-based representations naturally map variable-length inputs to a fixed dimensionality, raw time series require specific segmentation to ensure consistent input sizes. To address this without introducing zero-padding or truncation artifacts, we segment series into windows of fixed length $\ell$. In standard cases, these windows are non-overlapping. However, when the time series length $L$ is not a multiple of $\ell$, we introduce a partial overlap between the first two windows only. This adjustment shifts the segmentation grid of the subsequent windows such that the final window aligns exactly with the last time point of the series, preserving data integrity at the boundaries.

## A.6  DO SUPERVISED ENSEMBLES REALLY ENSEMBLE? (PART 2)

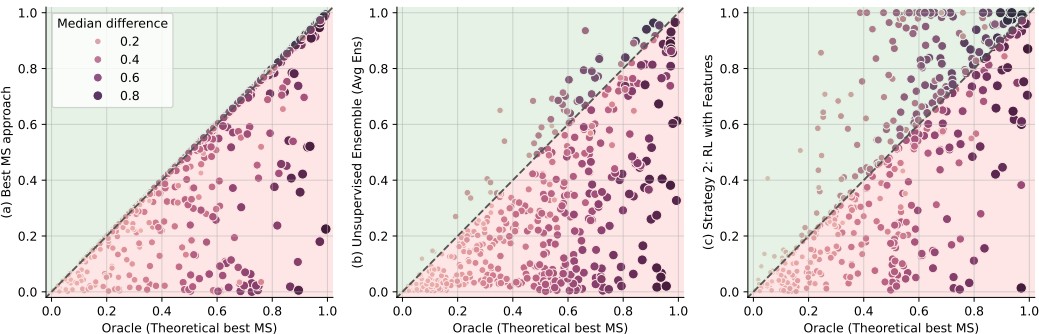

Figure 7: (left) Best model selection (Sylligardos et al., 2023), (middle) Unsupervised Ensemble, (right) Strategy 2 (with features) against the *Oracle*. For every time series, we represent the difference between the first and the median of all other detectors with the disk size and color.

In Figure 7, we focus specifically on time series that outperform the *Oracle*. First, when considering the three plots of Figure 7, we observe that small, clear disks are more often on the left. Big, dark disks are almost always on the right, because it is rare for the first two detectors to perform very well (small disk on the right), and impossible for very well-performing detectors to coexist with a low value of *Oracle*.

Overall, Figure 7 shows that time series with no single best detector are leading to poor ensemble performance (both unsupervised and supervised). Most of the time series for which the supervised ensembling approach outperforms the *Oracle* are associated with a strong single best detector.

## A.7 POINT ANOMALIES VERSUS SEQUENCE ANOMALIES

Figure 8 depicts a per-time series comparison between (left) Best model selection (Sylligardos et al., 2023), (middle) Unsupervised Ensemble, (right) Strategy 2 (with features) against the *Oracle*. Figure 8 focuses explicitly on the impact of anomaly types on the performance of the supervised ensembling approach (Strategy 2).

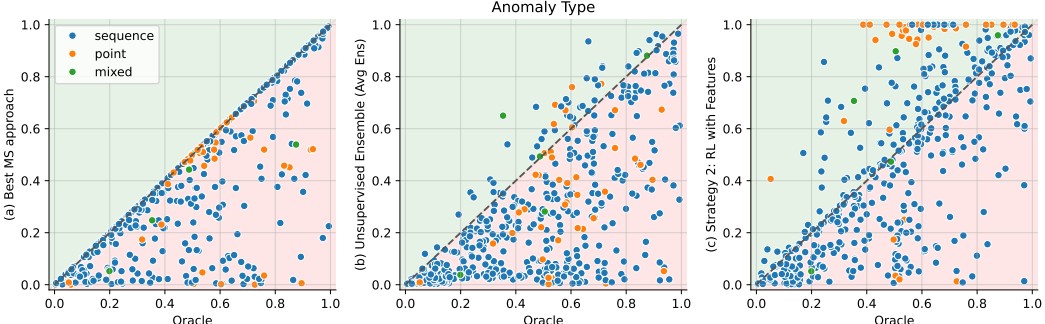

Figure 8: (left) Best model selection (Sylligardos et al., 2023), (middle) Unsupervised Ensemble, (right) Strategy 2 (with features) against the *Oracle*. For every time series, we represent the type of anomalies it contains with a specific colour.

## A.8 CRITICAL DIFFERENCE ANALYSIS

Figure 9 presents the results of the critical difference (CD) diagram, computed using pairwise Wilcoxon signed-rank tests. This analysis was conducted in the *in-distribution* setting to determine whether the performance differences between models are statistically significant.

From the diagram, it is evident that among the proposed models, only the RL-based strategies exhibit a statistically significant improvement compared to the other approaches. In contrast, the ML- and GP-based models show no significant difference in this evaluation, indicating comparable performance across the considered datasets.

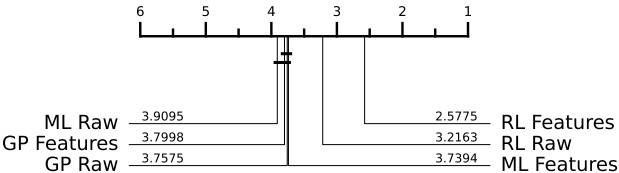

Figure 9: Critical difference diagram using the Wilcoxon signed rank test between strategies. Models connected by a horizontal line are not significantly different according to pairwise Wilcoxon signed-rank tests. RL-based strategies exhibit a significant performance difference compared to the other models.

## A.9 IN-DISTRIBUTION ACCURACY EVALUATION

Figure 10 displays an alternative form of the results illustrated in 4. In addition to the median and distribution of performance, this Figure also shows the mean performance of the strategies. Figure 11 displays the same type of information, but with the ML and GP strategies producing weights in $[-1, 1]$ while still in $[0, 1]$ for RL, as it is unstable otherwise. This instability is likely due to the variance introduced in the reward signal when subtracting detector outputs is allowed, destabilizing the policy gradient updates. These two plots in the *in-distribution* setting show that clipping yields better results. The clipped weights can sometimes be beneficial for the ensembling,

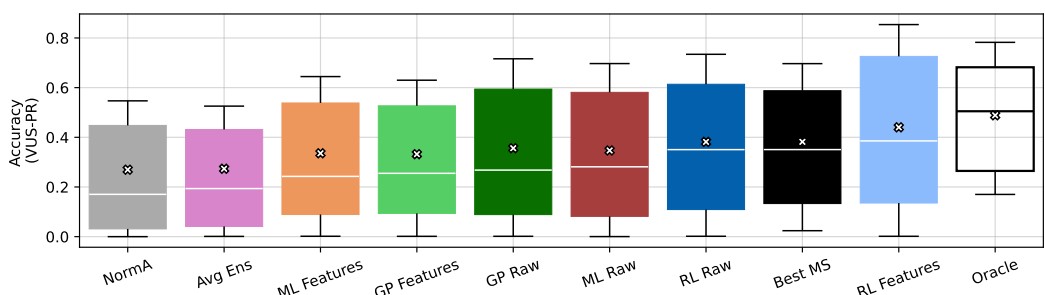

Figure 10: Overall accuracy averaged across 5 random seeds of all strategies, *Best MS*, *Avg Ens*, and *Oracle* in the *in-distribution* case with weights in $[0, 1]$ for all strategies.

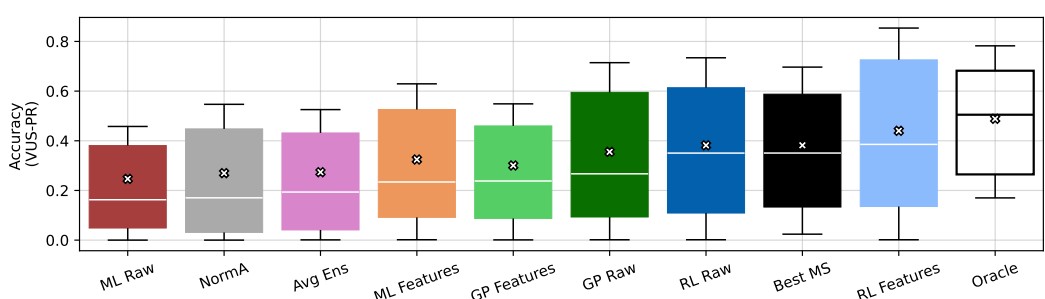

Figure 11: Overall accuracy averaged across 5 random seeds of all strategies, *Best MS*, *Avg Ens*, and *Oracle* in the *in-distribution* case with weights clipped in $[-1, 1]$ for ML and GP and clipped in $[0, 1]$ for RL for stability purposes.

but overall, they remove the poor-performing detectors, leading to better performances as illustrated in 12.

## A.10 OUT-OF-DISTRIBUTION ACCURACY EVALUATION

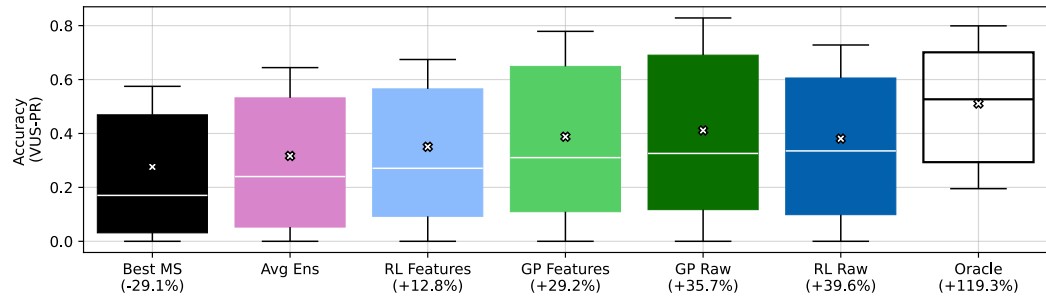

Figure 12: Overall accuracy of all strategies, *Best MS*, *Avg Ens*, and *Oracle* in the *out-of-distribution* case with weights clipped in $[0, 1]$ for all strategies.

Figure 12 displays an alternative form of the results illustrated in 4. In addition to the median and distribution of the performances, this plot also shows the percentage by which a given strategy outperforms the *Avg Ens* method. Figure 13 displays the same type of information, but with the GP approach producing weights in $[-1, 1]$. These two plots in the *out-of-distribution* setting demonstrate the benefits of clipping. We might have thought that the sparsity induced by the clipping would lead to weak performance in the *out-of-distribution* scenario, as we saw in Section 4.4 that denser activations perform better. However, clipping is moderated enough not to introduce to much sparsity in this case and improve the results as illustrated by Figure 12

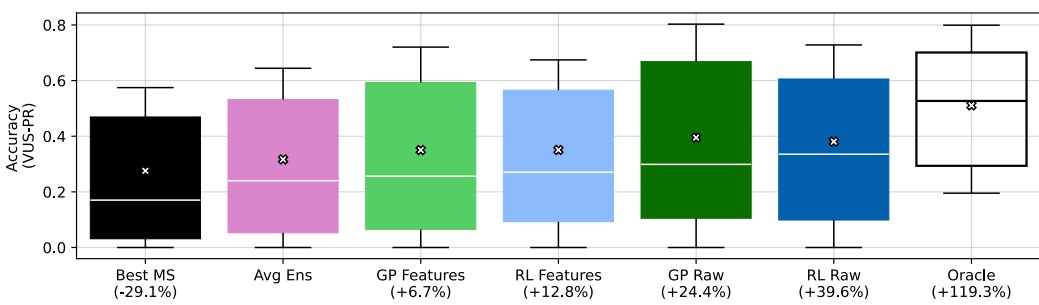

Figure 13: Overall accuracy of all strategies, *Best MS*, *Avg Ens*, and *Oracle* in the *out-of-distribution* case with weights in $[-1, 1]$ for GP and clipped in $[0, 1]$ for RL for stability purpose.

Figure 14 depicts the detailed results of the experiment on the *out-of-distribution* case, whose aggregated results are displayed in Figure 4. For a given dataset, the number of points per box plot is the number of series in the dataset. We observe that the *Oracle* is almost always more accurate than the ensembling and selecting methods, except for ECG and Occupancy, where the RL on features supervised ensembling is better. This is also the case for the YAHOO dataset, where the GP strategy applied to both features and raw data achieves higher accuracy.

For the sake of transparency, we also represent in Figure 15 the analogue of Figure 14 with the weights clipped in $[0, 1]$ for all strategies.

### A.11    WEIGHT DISTRIBUTION PER MODEL

Figure 16 shows the distribution of detector weights assigned by each model in the *in-distribution* setting; the clipped version of this plot sets all negative weights to zero. The ML- and GP-based strategies assign both positive and negative weights, whereas the RL-based models exclusively produce positive weights.

We also observe that some strategies consistently exclude specific detectors. For example, in the ML Features model, the LSTM and POLY detectors consistently receive zero weight, while the RL Raw model excludes four detectors entirely.

Finally, the spread of the weight distributions differs across strategies. The GP Raw model exhibits very compact distributions, indicating more stable weight assignments, while the ML Raw model shows the widest variability, suggesting greater sensitivity to different time series.

### A.12    T-SNE VISUALIZATIONS OF THE WEIGHTS.

Figure 17 aims at detecting whether all time series best analysed by a detector are similarly weighted by the different supervised ensembling strategies. We see, in the first place, that RL on features and ML on features tend to group the anomalies best detected by Norma (the best Individual detector). Moreover, the RL strategy based on features provides the most effective separation of the data into a compact representation from this perspective. Concerning the other methods, we observe that ML on raw data and GP on raw data both tend to gather some (but not all) time series best analyzed by IFOREST1 (in purple) and HBOS (in green) into distinct small clusters. These plots were computed with perplexity equals to 30 and without the use of Principal Component Analysis (Abdi & Williams, 2010).

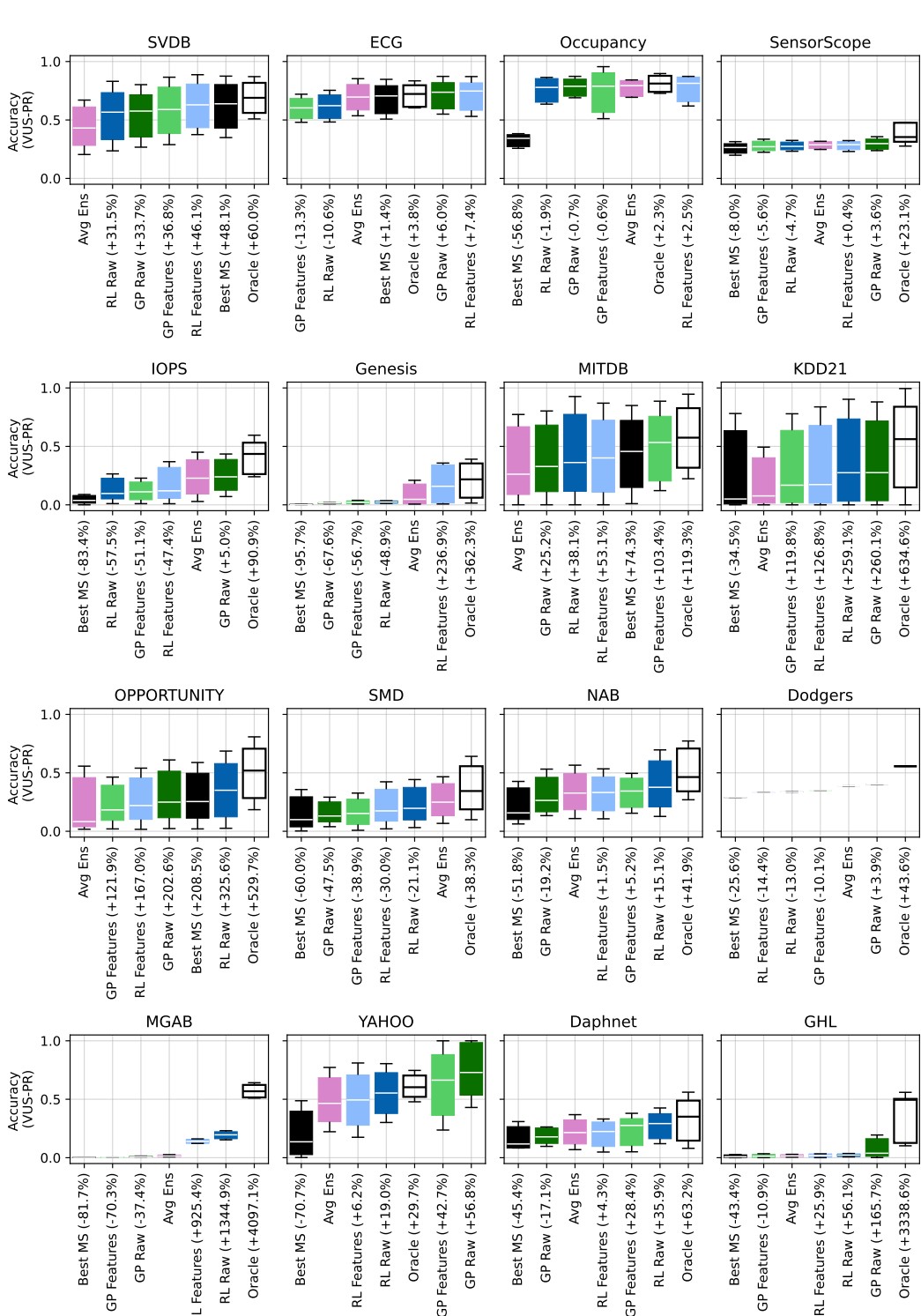

Figure 14: Per-dataset accuracy of all strategies, *Best MS*, *Avg Ens*, and *Oracle* in the *out-of-distribution* case with weights in $[-1, 1]$ for GP and clipped in $[0, 1]$ for RL for stability purpose.

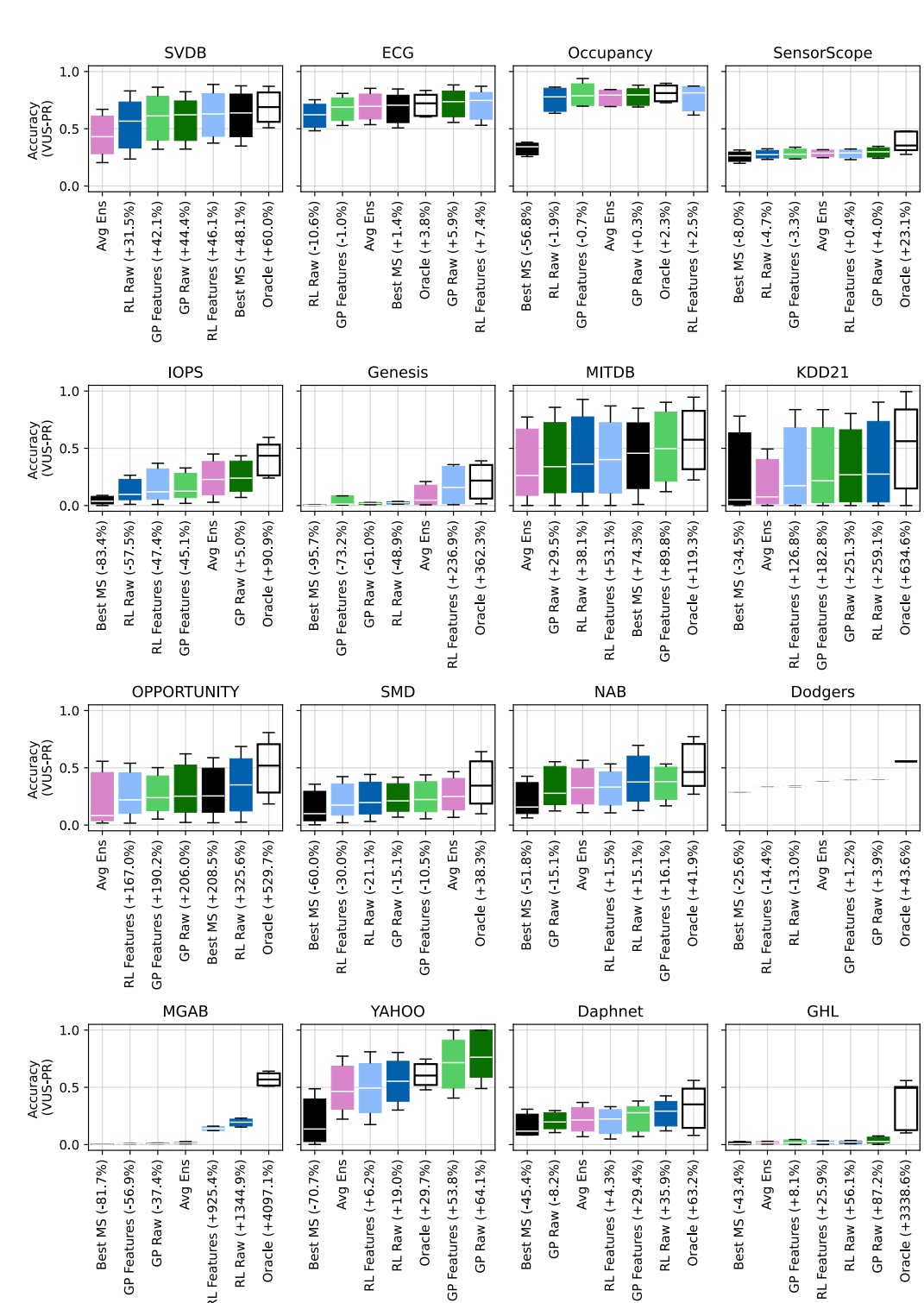

Figure 15: Per-dataset accuracy of all strategies, *Best MS*, *Avg Ens*, and *Oracle* in the *out-of-distribution* case with all strategies weights clipped in [0, 1]

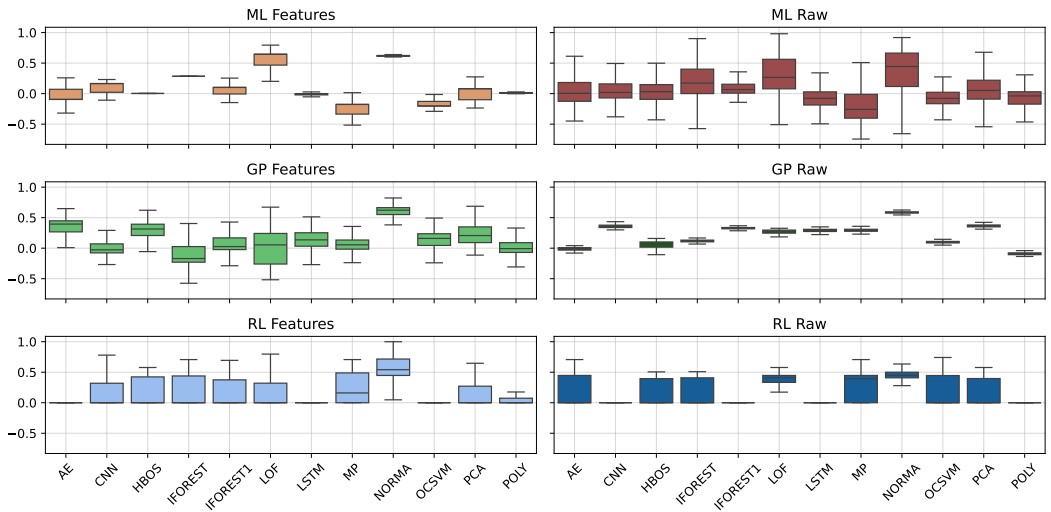

Figure 16: Weight distribution of detectors across different models. Each boxplot represents the variability of weights assigned to detectors by a given model in the *in-distribution* setting.

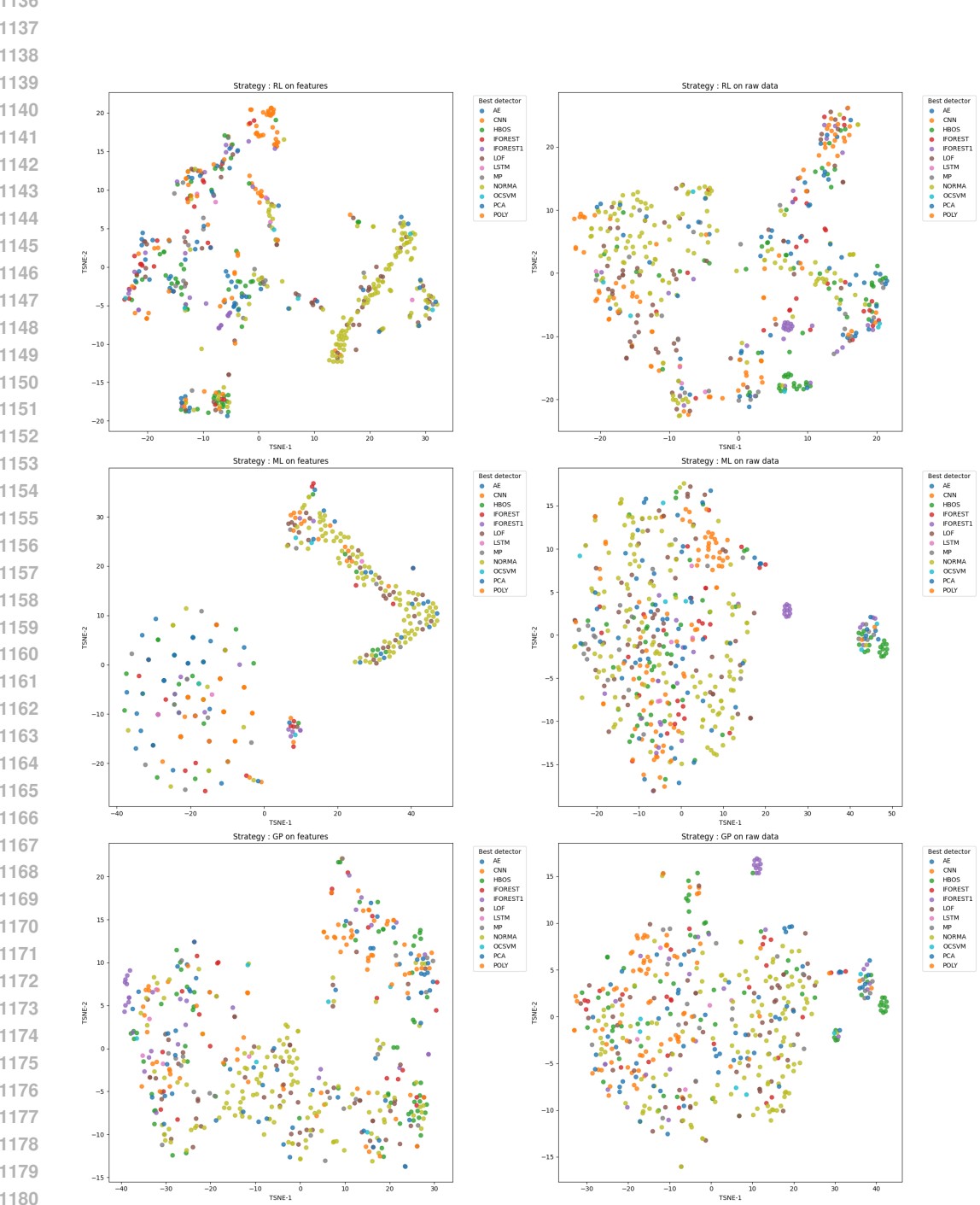

Figure 17: Per strategies 2D t-SNE visualisation of the weights for each time series labelled according to the best individual detector.

