# OpenReview forum: "Train, Mutate, or Reward? A Unified View of Supervised Ensembling for Time Series Anomaly Detection."
_ICLR.cc/2026/Conference — Submitted to ICLR 2026_

### Official Review · Reviewer_X87h · 2025-10-22

**Soundness:** 3
**Presentation:** 3
**Contribution:** 2
**Rating:** 4
**Confidence:** 4

**Summary:**

This paper studies supervised ensemble learning for time-series anomaly detection (TSAD), proposing a unified framework that formulates ensembling as a supervised problem. Three learning paradigms are examined, which are classical machine learning (ML), reinforcement learning (RL), and genetic programming (GP).  Each is trained to learn weights for combining anomaly detectors. The paper aims to evaluate these strategies under a unified experimental protocol on the TSB-UAD benchmark, showing that supervised ensembles can outperform both individual detectors and unsupervised ensemble methods.

**Strengths:**

1. The paper provides a well-structured overview of TSAD and positions supervised ensembling as a complementary paradigm to model selection and unsupervised ensembles.

2. The effort to evaluate three distinct strategies (,i.e., ML, RL, GP) within a controlled experimental pipeline is valuable for benchmarking purposes.

3. The paper is well-written and easy to follow. The research question itself is interesting.

**Weaknesses:**

1. The framework assumes access to labeled anomaly data for supervised training, which contradicts the fundamental challenge of anomaly detection (scarcity of labeled anomalies). The paper does not justify how this setup could be applied or adapted in realistic scenarios where labels are limited or unavailable.

2. The core idea, i.e., learning adaptive weights for multiple detectors, is conceptually close to mixture-of-experts (MoE) and existing adaptive weighting schemes. RL and GP are treated as alternative optimization strategies, not introducing fundamentally new principles for ensembling.

3. Only MLPs and small CNNs are used as the main models. More modern time-series architectures (Transformers, LSTMs, or xLSTMs) should be included to evaluate robustness across model.

4. No ablation studies on model depth, hierarchy, or the number of detectors are provided to verify stability of the observations.

5. The analysis remains largely empirical. The authors report that one method outperforms another, but do not explain why RL succeeds or how the learned weight distribution differs from ML or GP.

6. Qualitative results (e.g., visualization of weight distributions, failure cases, or t-SNE plots of feature spaces) would strengthen the interpretability of findings.

7. The benchmark lacks comparison to recent state-of-the-art TSAD models (e.g., TranAD, OmniAnomaly, Anomaly Transformer,  etc.). Without these, it is unclear whether supervised ensembling truly advances the field.

8. Figures (e.g., Fig. 4) lack explanations of error bars and variance computation.

9. Quantitative results are mostly presented in plots; tabulated performance scores would improve clarity.

**Questions:**

1. How do the authors justify the assumption of having sufficient labeled anomaly data for supervised training, given that real-world TSAD scenarios are typically label-scarce?

2. Could the proposed framework be extended or adapted to semi-supervised or weakly-supervised settings?

3. The proposed supervised ensemble approach appears conceptually close to mixture-of-experts (MoE) or adaptive weighting frameworks. Could the authors elaborate on what fundamentally differentiates their method from MoE-based approaches?

4. Why should RL or GP be viewed as conceptually distinct ensemble learning paradigms rather than merely alternative optimization schemes?

5. Why did the authors restrict the core models to simple MLPs and CNNs?

6. Have the authors explored more modern architectures such as Transformers, LSTMs, or xLSTMs? If not, could the authors discuss expected differences or challenges in incorporating these architectures?

7. Could the authors provide ablation results on model depth, architectural hierarchy, and the number of detectors to assess the stability of the conclusions?

8. How sensitive are the main findings (e.g., RL outperforming ML/GP) to these hyperparameters?

9. What are the theoretical or intuitive reasons that explain why RL-based ensembling performs better than ML or GP in certain settings?

10. How do the learned weight distributions differ across the three strategies (ML, RL, GP)?

11. Could the authors provide qualitative analyses such as t-SNE visualizations, detector weight distributions, or case studies of failure modes to better understand model behavior?

12. Why were recent state-of-the-art TSAD methods (e.g., TranAD, OmniAnomaly, Anomaly Transformer, THOC, etc.) excluded from the benchmark? Would the proposed supervised ensemble still outperform these modern baselines?

13. In Figure 4, what do the error bars represent, and how were they computed (e.g., across runs, datasets, or random seeds)? Could the authors provide the variance or confidence intervals explicitly?

14. Could the authors include tabulated performance scores (e.g., mean ± std) for all methods across datasets to improve transparency and enable easier cross-paper comparison?

---

> ### Author Response · Authors · 2025-11-22
>
> > How do the authors justify the assumption of having sufficient labeled anomaly data for supervised training, given that real-world TSAD scenarios are typically label-scarce?
>
> We thank the reviewer for raising this crucial point regarding the practicality of supervised approaches given the scarcity of labeled anomalies. We fully acknowledge that labeled anomalies are rare in real-world deployments, which has historically justified the field's focus on unsupervised methods.
>
> **Our goal is not to assume abundant labeled anomalies are available, but rather to investigate whether limited supervision can be leveraged to surpass the performance ceiling of unsupervised methods.**
>
> Recent benchmarks [1, 2, 3] highlight that individual detectors and unsupervised selection methods often hit a performance plateau. In addition to the 3 questions raised in the main text of the article, our work also addresses the question “Can we use labels from historical or related datasets to learn robust combination rules that generalize to new, unlabeled contexts ?” (see out-of-distribution experiments in Figure 4).
>
> To directly address the constraint of label scarcity, we explicitly designed our out-of-distribution (OOD) experiments (Leave-One-Dataset-Out). The ensemble is trained on multiple datasets and evaluated on a completely held-out dataset with distinct anomaly characteristics. This setup mirrors realistic conditions where labels might be available for some historical processes but are unavailable, sparse, or mismatched for the deployment target.
>
> The results of these OOD experiments are critical. **We observe that our supervised ensembles retain competitive performance against model selection even when applied to domains they were not trained on.**This suggests that supervision allows the model (especially the RL agent) to learn how to combine detectors rather than memorizing specific anomalies.
>
> We clarify this connection in the revised manuscript, explicitly stating that the OOD evaluations serve as a proxy for robustness assessment in label-scarce scenarios and demonstrating that supervision can be transferable and practically valuable.
>
> References:
>
> [1] Sylligardos, G., et al. (2023). Choose Wisely: An Extensive Evaluation of Model Selection for Anomaly Detection in Time Series.
>
> [2] Paparrizos, J., et al. (2022). TSB-UAD: An End-to-End Benchmark Suite for Univariate Time-Series Anomaly Detection. PVLDB.
>
> [3] Alpogon, et al. (2024). AutoTSAD. (Associated with automated benchmark studies).
>
> **Overall, thanks to this comment, we applied the following modification to the paper:**
>
> - We have added in the introduction a passage insisting on the motivation : “We emphasize here that we do not assume that abundant labeled anomalies are available in general but rather investigate whether limited supervision can be
> leveraged to surpass the performance ceiling of unsupervised methods.”
> - We have added in section 4.2 a passage on  the value of OOD experiments : "[...] we evaluate the performance of the
> models in an out-of-distribution setting (i.e., evaluated on a dataset not used in the training set) serving as a proxy for robustness assessment in label-scarce scenarios and demonstrating that supervision can be transferable and practically valuable."

---

> ### Author Response · Authors · 2025-11-22
>
> > Could the proposed framework be extended or adapted to semi-supervised or weakly-supervised settings?
>
> We thank the reviewer for this forward-looking question. While our current work focuses on the fully supervised setting to establish a strong upper-bound baseline, we believe our RL and GP frameworks are well-suited for adaptation to label-scarce regimes.
>
> **Before discussing future adaptations, we wish to highlight the robustness of our current evaluation.** We consider the VUS-PR metric [1] in all our experiments. As detailed in [1], this metric is specifically designed to account for imprecise, noisy, misaligned, or time-delayed labels, ensuring that our reported performance is robust to the inherent ambiguity of anomaly timestamps.
>
> **In weakly supervised scenarios [2] where only coarse-grained labels are available** knowing that a window contains an anomaly without exact timestamps), our RL/GP frameworks offer a distinct advantage over standard gradient-based ML methods. One can design a non-differentiable reward/fitness function tailored to the label structure. For instance, a discrete step function that yields a positive reward for a "True Positive" at the window level and a negative reward otherwise. Such non-differentiable objectives cannot be directly optimized by standard gradient-based methods (such as the ML baseline), but are naturally handled by RL (via policy gradient) or GP optimization.
>
> **In semi-supervised settings where labeled anomalies are scarce, the training pipeline could be extended.** The RL agent or GP model could be trained on the labeled set while considering a consistency-based auxiliary loss (inspired by [3]) on the unlabeled data. This would force the model to yield consistent predictions under perturbations.
>
> However, we approach this extension with caution.**As noted in surveys on semi-supervised learning [4], introducing unlabeled data does not guarantee improvement and can sometimes degrade performance.** Therefore, while semi-supervised learning is a promising research direction for identifying an effective strategy, it introduces complexity regarding "safe" learning, which **falls outside the scope of this initial baseline study.**
>
> References:
>
> [1] Boniol, P., et al. (2025). VUS: effective and efficient accuracy measures for time-series anomaly detection. The VLDB Journal, 34(3), 32.
>
> [2] Zhou, Z. H. (2018). A brief introduction to weakly supervised learning. National Science Review, 5(1), 44-53.
>
> [3] Zhang, X., Zhao, Z., Tsiligkaridis, T., & Zitnik, M. (2022). Self-supervised contrastive pre-training for time series via time-frequency consistency. Advances in neural information processing systems, 35, 3988-4003.
>
> [4] Van Engelen, J. E., & Hoos, H. H. (2020). A survey on semi-supervised learning. Machine Learning, 109(2), 373-440.
>
> **Overall, thanks to this comment, we applied the following modification to the paper:**
> - We have updated the paragraph dedicated to the evaluation metric in the Appendix A.5 to insist on the robustness of VUS-PR in weakly supervised settings : "As detailed in Boniol et al. (2025), this metric is specifically designed to account
> for imprecise, noisy, misaligned or time-delayed labels, ensuring that our reported performance is
> robust to the inherent ambiguity of anomaly timestamps."

---

> ### Author Response · Authors · 2025-11-22
>
> > The proposed supervised ensemble approach appears conceptually close to mixture-of-experts (MoE) or adaptive weighting frameworks. Could the authors elaborate on what fundamentally differentiates their method from MoE-based approaches?
>
> We thank the reviewer for this insightful comment. We agree that our framework is structurally analogous to a Mixture-of-Experts (MoE) or adaptive weighting system, where the "experts" correspond to the individual anomaly detectors and the "gating network" corresponds to our aggregation model (MLP or CNN).
>
> However, our work differentiates itself from standard MoE implementations in two fundamental ways, driven by the unique challenges of Time Series Anomaly Detection (TSAD).
>
> Standard MoE frameworks typically rely on differentiable loss functions (such as Cross-Entropy or MSE) to train the gating network via backpropagation (see Section D of [1]). In contrast, our RL and GP approaches are specifically designed to optimize non-differentiable metrics tailored for TSAD (such as VUS-PR).
>
> While our Strategy 1 (ML) is closest to a standard MoE approach using a differentiable proxy (MSE), we empirically demonstrate that it is often suboptimal. Our Strategies 2 (RL) and 3 (GP) bypass this limitation, allowing the "gating network" to optimize the true TSAD metric directly.
>
> A classic MoE typically involves trainable neural network experts, co-optimized with the gating network. The "expert" layers and the "gating" layers learn simultaneously. **Contrary to MoE in our framework, the experts are a diverse set of pre-existing, heterogeneous, and fixed algorithms (e.g., LOF [3], Isolation Forest [4], NormA [2]). These are often unsupervised.**
>
> Our framework adapts the MoE concept to a setting where the experts are frozen. This shifts the complexity entirely to the learned
> aggregation policy, rather than distributing it across the training of experts.
>
> References:
>
> [1] Mu, S., & Lin, S. (2025). A comprehensive survey of mixture-of-experts: Algorithms, theory, and applications. arXiv preprint arXiv:2503.07137.
>
> [2] Boniol, P., et al. (2021). Unsupervised and scalable subsequence anomaly detection in large data series. The VLDB Journal.
>
> [3] Breunig, M. M., et al. (2000). LOF: Identifying Density-based Local Outliers. ACM SIGMOD.
>
> [4] Liu, F. T., Ting, K. M., & Zhou, Z. H. (2008). Isolation Forest. IEEE ICDM.
>
> **Overall, thanks to this comment, we applied the following modification to the paper :**
> - We have updated Section 3.1 on the description of our pipeline by adding a seentence on the difference with MoE : "Contrary to Mixture of Experts (MoE) (Yuksel et al., 2012) in our framework, the experts are a diverse set of pre-existing, heterogeneous, and pre-trained algorithms."

---

> ### Author Response · Authors · 2025-11-22
>
> > Why should RL or GP be viewed as conceptually distinct ensemble learning paradigms rather than merely alternative optimization schemes?
>
> Thank you for this question on the fundamental differences between GP and RL. We agree that RL and GP are different optimization tools. However, we find **the distinction into 'paradigms' useful to characterize their fundamentally different mechanisms of exploration, which directly impacts their performance in our experiments.**
>
> **The distinction lies in how the ensemble task is modeled.**
>
> RL frames ensembling as a control problem [1]. The ensemble is modeled as an agent that interacts with the data (state), takes actions (weighting), and updates its policy via dynamic gradient estimation (as in PPO [2]) based on reward feedback. **It optimizes a decision-making trajectory with an adaptive balance between exploration and exploitation.**
>
> GP frames ensembling as a global search problem [3]. Instead of estimating gradients or optimizing a single policy, **it maintains a population of diverse solutions.** It relies on evolutionary operators (mutation and crossover) to explore the parameter space, maintaining **a fixed balance between exploration and exploitation** without requiring differentiability.
>
> **This distinction is further supported by the literature on hybrid frameworks, such as Evolutionary Algorithm Reinforcement Learning (EARL) [4]**. These frameworks leverage the fact that the two methods address different aspects of the optimization landscape: GP is often used to navigate the topology (Global Search), while RL is employed to fine-tune via gradient ascent (Exploitation).
>
> To illustrate this complementarity, a typical EARL process functions as follows:
> - Initialize a population of policies (MLPs).
> - Evaluate all policies to obtain rewards.
> - Select top-performing individuals (Evolutionary step).
> - Update these individuals using PPO to maximize local performance (RL step).
> - Mutate/Crossover to generate new diverse policies (Evolutionary step).
>
> This ability to offset each other's limitations further illustrates the utility of the 'paradigm' distinction in capturing their fundamentally different exploration mechanisms.
>
> References:
>
> [1] Sutton, R. S., & Barto, A. G. (2018). Reinforcement learning: An introduction. MIT press.
>
> [2] Schulman, J., et al. (2017). Proximal policy optimization algorithms. arXiv preprint arXiv:1707.06347.
>
> [3] Mitchell, M. (1998). An introduction to genetic algorithms. MIT press.
>
> [4] Moriarty, D. E., Schultz, A. C., & Grefenstette, J. J. (1999). Evolutionary algorithms for reinforcement learning. Journal of Artificial Intelligence Research, 11, 241-276.
>
> **Overall, thanks to this comment, we applied the following modification to the paper :**
> - We added a sentence in section 3.4 on genetic programming to highlight how it can complement Reinforcement Learning through EARL : "This complementarity is further supported by the literature on hybrid frameworks, such as Evolutionary Algorithm Reinforcement Learning(EARL) (Moriarty et al., 1999)."

---

> ### Author Response · Authors · 2025-11-22
>
> > Why did the authors restrict the core models to simple MLPs and CNNs?
>
> > Have the authors explored more modern architectures such as Transformers, LSTMs, or xLSTMs? If not, could the authors discuss expected differences or challenges in incorporating these architectures? If not, could the authors discuss expected differences or challenges in incorporating these architectures?
>
> We thank the reviewer for raising these three questions regarding our architectural choices. It allows us to clarify the methodological scope of our baseline.
>
> **Our primary goal is to establish a clean, fair, and reproducible baseline for supervised ensembling in Time-Series Anomaly Detection (TSAD).** To ensure our results are strictly comparable to state-of-the-art approaches, we aligned our architectures with the strongest-performing **models identified in the comprehensive benchmark by [1].**
>
> In the task of supervised model selection (which is closely related to our weighting task), **MLP on features (catch22) and CNN on raw series were shown to outperform more complex architectures,** including Transformers, under identical experimental conditions. We intentionally restricted our study to these architectures to ensure our baseline reflects current "best practices" in the domain rather than introducing further complexity.
>
> **We aim to isolate the contribution of the supervised ensembling strategy (ML vs. RL vs. GP) itself.** Using simple yet powerful baselines (MLPs/CNNs) ensures that performance differences can be attributed to the learning paradigm rather than confounded by the architectural complexity or training instability often associated with Recurrent Neural Networks (LSTMs/xLSTMs).
>
> **While modern sequence models are powerful, benchmarking the possible architectures is outside the scope of this work, which focuses on establishing the first supervised ensemble baseline.**
>
> Regarding the specific suggestion of LSTMs, introducing LSTMs as the aggregation model would hinder fair comparison with the Model Selection baselines (which rely on MLP/CNN). We also note that temporal dynamics are not ignored: one of the 12 underlying detectors in our pool is an LSTM-AD. Interestingly, this detector does not consistently outperform others (such as the CNN or Auto-Encoder), further justifying our decision not to prioritize recurrence in the aggregation layer.
>
> That said, we agree that an additional comparison of the supervised ensembling strategy can be conducted, incorporating modern sequence models such as LSTMs or xLSTMs to examine the impact of model complexity on the strategy's performance. **However, our motivation is to restrict ourselves to a simple and controlled environment in which the learning paradigms can be compared.** Studying the impact of temporal sequence models within the model selection or ensembling framework is an interesting future research direction.
>
> Overall, for the scope of this paper, we think that it is more appropriate to prioritize consistency with the existing literature and reproducibility.
>
> References:
>
> [1] Sylligardos, G., et al. (2023). Choose Wisely: An Extensive Evaluation of Model Selection for Anomaly Detection in Time Series. (PVLDB).
>
> **Overall, thanks to this comment, we applied the following modification to the paper :**
> - We added an important sentence in the introduction detailings our motivations and the role of potential new architectures : "Crucially, our goal is not to compare architectures (CNN vs LSTMs) [...] but to benchmark our strategies upon a simple, reproducible baseline for supervised ensembling grounded in existing research. "
> - We modified Table 1 to emphasize that MLP and CNN were the best performing model selection methods in [1] respectively on features and raw data.

---

> ### Author Response · Authors · 2025-11-22
>
> > Could the authors provide ablation results on model depth, architectural hierarchy, and the number of detectors to assess the stability of the conclusions?
>
> > How sensitive are the main findings (e.g., RL outperforming ML/GP) to these hyperparameters?
>
> We thank the reviewer for these suggestions regarding ablations acroos the strategies on model depth, hierarchy, and detector count. While we agree that these are valuable analyses for optimizing a specific architecture, we respectfully submit that they fall outside the scope of this study.
>
> **Our primary goal is not to optimize the hyperparameters of the individual aggregation networks, but rather to evaluate and compare the supervised ensemble strategies themselves (ML vs. RL vs. GP) under strictly controlled conditions.**
> To isolate the effect of the learning paradigm, we deliberately fixed the core components: the backbone models (MLP/CNN), the feature representations (catch22), and the detector set across all methods.
>
> Simultaneously varying the model architecture or the detector pool would introduce additional confounding factors. This would make it difficult to disentangle whether a performance gain is due to the learning strategy or simply due to a specific architecture's capacity.
>
> As noted in our response regarding Model Selection baselines, our work is designed to measure strategy-level behavior rather than
> architecture sensitivity. We explicitly avoided fine-tuning the aggregator's complexity to prevent hyperparameter overfitting that might favor one strategy over another.
>
> To ensure the robustness of our conclusions without introducing architectural variables, **we focused our resources on stochastic stability analysis**. As added in the Appendix, we conducted stability experiments across five random seeds for all methods (including constrained/unconstrained variations for GP and ML). These experiments confirm that the relative performance ordering (RL outperforming ML/GP) is consistent. **We believe that robustness to training stochasticity is the primary prerequisite for a fair comparison of ML, RL, and GP, and that it takes precedence over architectural ablation in this foundational study.**
>
> Finally, we agree that a deeper exploration of architectural depth and detector subset influence is a direction for future work, particularly to understand how supervised ensembling scales with more complex models. **However, for this comparative and experimental evaluation paper, we prioritize experimental control and reproducibility.**
>
> **Overall, thanks to this comment, we applied the following modification to the paper :**
>
> -  We added an important paragraph in the introduction describing more precisely our motivations :  "Crucially, our goal is not to compare [...] hyperparameter sensitivity, but to benchmark our strategies upon a simple, reproducible baseline for supervised ensembling grounded in existing research."

---

> ### Author Response · Authors · 2025-11-22
>
> > What are the theoretical or intuitive reasons that explain why RL-based ensembling performs better than ML or GP in certain settings?
>
> We thank the reviewer for this insightful question. The observed performance hierarchy (RL > GP, ML) can be explained by analyzing two fundamental components: **the metric each method optimizes and the way it is optimized.**
>
> **The primary reason RL outperforms ML is the objective function itself.** Both paradigms (RL/GP) use the actual anomaly detection metric (VUS-PR) as a reward function (RL) or a fitness function (GP). **They directly maximize the quantity of interest.** In contrast, the classical **ML baseline relies on a differentiable proxy objective (MSE).** While MSE is standard for regression, it does not perfectly align with VUS-PR, explaining the lower performance of the ML baseline.
>
> **The superiority of RL over GP, despite both optimizing the true metric, stems from the efficiency of RL's search mechanism.** RL (via PPO) benefits from gradient-based guidance. By estimating the policy gradient, the agent can identify directionally meaningful updates to the weights. This allows for fine-grained adjustments and faster convergence toward a local optimum.
> Conversely, GP relies on a population-based stochastic search driven by random mutations and crossovers. **While this derivative-free approach is powerful for global exploration (avoiding local optima), it is inherently less efficient for fine-tuning.**It essentially "stumbles" upon improvements rather than computing the path toward them.
>
> In summary, **methods that directly optimize true anomaly-detection accuracy (RL and GP) reliably outperform those that rely on proxy objectives (ML).** Among the directed  methods, RL prevails because its guided optimization dynamics are more sample-efficient and precise than GP's stochastic search for this specific task.
>
> **Overall, thanks to this comment, we applied the following modification to the paper :**
> - We added an important sentence in section 4.2 on the reasons of the strong performance of RL : "This performance of Strategy 2 (RL) is likely due to its guided optimization on the true TSAD mertic VUS-PR."
>
> > How do the learned weight distributions differ across the three strategies (ML, RL, GP)?
>
> We thank the reviewer for this question, which allows us to deepen the analysis of the behavioral differences between the three strategies. **The optimization dynamics described in our previous response directly influence the sparsity and generalization properties of each method.**
>
> As previously mentioned, RL is a directed method (policy gradient). It uses gradient estimation to identify and exploit the most rewarding directions. As a consequence, **the agent selects the most adapted anomaly detectors, effectively removing less relevant ones.** As shown in Figure 6(c) of the main text, RL exhibits high sparsity (activates fewer detectors). This allows RL to achieve superior performance in in-distribution settings, as it effectively fits a performing combination for the known training distribution.
> On the other hand, GP performs a global exploration of the parameter space via random mutation and selection. Thus, it does not follow a gradient but maintains a population of diverse solutions. This stochastic process leads to slower convergence and tends to activate more detectors. **While this lack of specialization results in lower ID performance compared to RL, it becomes an advantage in Out-of-Distribution (OOD) settings. Because GP relies on a broader, more diverse set of detectors, it is less sensitive to distributional shifts. Our experiments confirm that RL is not uniformly superior: GP performs competitively and in OOD scenarios due to its inherent robustness.**
>
> Finally, regarding the ML baseline, since it is also a gradient-based method (optimizing MSE), it shares the sparsity characteristic of RL. As observed in Figure 6(c), ML tends to activate fewer detectors than GP.
>
> **Overall, thanks to this comment, we applied the following modification to the paper :**
> - We added a sentence in Section 4.4 pointing out that the more directed (gradient-based) the method is, the fewer activated detectors : "We also make the observation that gradient based methods like ML of RL (with
> PPO) induce sparser models activations."

---

> ### Author Response · Authors · 2025-11-22
>
> > Could the authors provide qualitative analyses such as t-SNE visualizations, detector weight distributions, or case studies of failure modes to better understand model behavior?
>
> Thanks to this comment, we applied the following modification to the paper :
> - We created a section A.12 on T-SNE visualisations to address the demand. We focused the analysis of the T-SNE plots by labelling the time series with their best detector to determine whether the supervised ensemble groups time series based on their best detector.
>
> > Why were recent state-of-the-art TSAD methods (e.g., TranAD, OmniAnomaly, Anomaly Transformer, THOC, etc.) excluded from the benchmark? Would the proposed supervised ensemble still outperform these modern baselines?
>
> We thank the reviewer for suggesting the inclusion of recent state-of-the-art methods such as TranAD [1], OmniAnomaly, and Anomaly Transformer. We agree that the landscape of TSAD is evolving rapidly. However, our decision to fix the current detector pool was driven by a specific methodological goal, grounded in the findings of the most recent benchmarks.
> **Recent extensive evaluation studies, such as TSB-AD (NeurIPS 2024) [2] and TAB (VLDB 2025) [3], have demonstrated a crucial reality: there is no single "best" method overall.**
>
> Even sophisticated deep learning models (such as TranAD, MOMENT [4], or Anomaly Transformer [5]) exhibit high performance variance, excelling on some datasets while failing on others, whereas simpler methods (such as Isolation Forest or Matrix Profile) perform better.
>
> **As a consequence, replacing our pool with more recent methods would not change the fundamental problem. The need remains to dynamically select and combine the right tools for the right data.**
>
> **Our primary contribution is to identify the most appropriate learning principles (RL, GP, or ML) for supervised ensembling, rather than benchmarking individual detectors.** The specific composition of the pool serves as a proxy for a "set of heterogeneous experts." We hypothesize that **changing the pool (e.g., adding TranAD) would likely improve the absolute performance of our ensemble** (as the RL agent would have access to stronger experts), **but it would not alter the relative conclusions regarding the learning strategies.**
>
> Finally, we state that our supervised ensembling framework is model-agnostic: it is designed to integrate these modern baselines as they emerge, leveraging their strengths whenever they are locally optimal.
>
> References:
>
> [1] Tuli, S., Casale, G., & Jennings, N. R. (2022). Tranad: Deep transformer networks for anomaly detection in multivariate time series data. arXiv preprint arXiv:2201.07284.
>
> [2] Paparrizos, J., et al. (2024). TSB-AD: The Time Series Benchmark for Anomaly Detection. NeurIPS 2024.
>
> [3] Boniol, P., et al. (2025). TAB: The Time Series Anomaly Detection Benchmark. VLDB 2025.
>
> [4] Goswami, M., Szafer, K., Choudhry, A., Cai, Y., Li, S., & Dubrawski, A. (2024). Moment: A family of open time-series foundation models. arXiv preprint arXiv:2402.03885.
>
> [5] Xu, J., Wu, H., Wang, J., & Long, M. (2021). Anomaly transformer: Time series anomaly detection with association discrepancy. arXiv preprint arXiv:2110.02642.
>
> **Overall, thanks to this comment, we applied the following modification to the paper** :
> - We added an important paragraph in the introduction describing more precisely our motivations :  "Crucially, our goal is not to [...] change the detector pool but to benchmark our strategies upon a simple, reproducible baseline for supervised ensembling grounded in existing research."
>
> > In Figure 4, what do the error bars represent, and how were they computed (e.g., across runs, datasets, or random seeds)? Could the authors provide the variance or confidence intervals explicitly?
>
> **Overall, thanks to this comment, we applied the following modification to the paper** :
> - We changed the caption of Figure 4 by adding the description of the error bars and how they were computed : "Figure 4: [...] The box extends from the first quartile (Q1) to the third quartile (Q3) with the median line. Whiskers cover the data range up to 0.24 × (Q3-Q1) from the box edges."
>
> > Could the authors include tabulated performance scores (e.g., mean ± std) for all methods across datasets to improve transparency and enable easier cross-paper comparison?
>
> **Overall, thanks to this comment, we applied the following modification to the paper** :
> - We added table 2 in the main text assessing the variability of all the strategies. The caption reads "Variability study of the mean performance of each strategy across 5 random seeds in the in-distribution settings (mean ± std). Strategy 2 (RL) on features consistently outperforms the other approaches."

---

> > ### Comment · Reviewer_X87h · 2025-11-27
> > **Response to the authors**
> >
> > Thank you very much for your detailed response. I have carefully checked the rebuttal and the paper, and my concerns have been well addressed.
> >
> > I have decided to raise my score to **8**.
> >
> > Good luck with your submission.

---

### Official Review · Reviewer_vtUc · 2025-11-01

**Soundness:** 2
**Presentation:** 3
**Contribution:** 2
**Rating:** 4
**Confidence:** 3

**Summary:**

For time-series anomaly detection, the authors propose supervised ensemble learning.   The ensemble of detectors are linearly weighted and combined.  Three strategies were considered as the underlying detectors: gradient based machine learning (ML) method, reinforcement learning (RL) and genetic programming (GP).  For the gradient-based method, they use MSE as the loss function.  For the reinforcement learning method, they use Proximal Policy Optimization (PPO) algorithm.  For genetic algorithms, each hypothesis/individual in the population is a neural network and the fitness function is accuracy.

The baseline methods are 12 individual detections from the TSB-UAD benchmark, unsupervised ensemble (average of the output of  individual detectors), the best method in (Sylligardos et al., 2023).  They have two scenarios: in-distribution (all 16 datasets from TSB-UAD) in training and test sets) and out-of-distribution.(15 datasets for training, one for testing).  For in-distribution, they found the supervised ensemble with RL has a better performance.  For out-of-distribution, ensemble with RL and GP have a better performance.  When compared the Oracle (the best model for a time series), Supervised ensembles using RL with features can outperform the oracle on some time series.   For in-distribution, they find fewer detectors better, while for out-of-distribution, more detectors better.

**Strengths:**

1.  Exploring ensemble methods with 3 strategies (ML, RL, and GP) is interesting.

2. The empirical results indicate that supervised ensemble using RL with statistical features generally outperform the compared methods.

3.  The paper is generally well organized and written.

**Weaknesses:**

1.  Most of the proposed methods are existing so the novelty level is not high.

2.  Other ensemble methods were not included for comparison.

**Questions:**

Sec 3.1: what are the statistical features?

Sec 4.4: Any insights on why more activated detectors are better in out-of-distributions, while fewer activated detectors are better in in-distribution?

Comments:

Figure 5, x axis: Oralee -> Oracle

---

> ### Author Response · Authors · 2025-11-22
>
> >Most of the proposed methods are existing so the novelty level is not high.
>
> We thank the reviewer for questioning the motivations and the novelty of our study. We agree that the individual algorithmic components we use (e.g., PPO for RL, Genetic Algorithms for GP, Adam optimizer for ML) are well-established in the broader machine learning community. However, the contribution and novelty are elsewhere.
>
> **Our contribution does not lie in inventing a new optimization algorithm, but rather in introducing the first systematic formulation and evaluation of Supervised Ensembling for Time-Series Anomaly Detection (TSAD).**
>
> The Time Series Anomaly Detection (TSAD) field has heavily relied on unsupervised ensembling (simple averaging) or model selection. In our case, we propose the first unified framework that treats TSAD ensembling as a supervised learning problem across three distinct paradigms (ML, RL, GP). **We demonstrate how to adapt these paradigms to the specific constraints of TSAD: optimizing non-differentiable metrics (VUS-PR) and aggregating fixed, heterogeneous experts without retraining them.**
>
> The Scientific novelty lies in the discovery of relationships between methods. **We reveal that the choice of learning strategy is more significant than the choice of data representation.**
>
> Finally, **our goal is to establish a rigorous baseline**. However, a robust baseline must rely on established, reproducible building blocks rather than highly complex custom architectures. **By showing that standard RL (PPO) can beat state-of-the-art model selection methods in TSAD, we provide a clear, reproducible starting point for future research in supervised ensembling.**
> To conclude, our **novelty lies in the synthesis and rigorous benchmarking of these methods in a new domain (Supervised TSAD Ensembling)**, filling a critical gap identified in recent literature.
>
> **Overall, thanks to this comment, we applied the following modification to the paper:**
> - We added the following two references in Section 2.3 : "While unsupervised ensembling offers simplicity (Goswami et al.,
> 2022; Schmidl et al., 2024) and robustness [...]" to  to highlight the existing literature context on unsupervised ensembling and thus the novelty of supervised ensembling.
>
> > Other ensemble methods were not included for comparison.
>
> We thank the reviewer for this comment regarding the inclusion of other ensemble methods in our study.
>
> In this work, we introduce, to the best of our knowledge, the first baseline for supervised ensembling specifically tailored to Time Series Anomaly Detection (TSAD). **Our primary goal is not to perform an exhaustive architectural benchmark of all existing ensemble techniques (such as Stacking, Boosting, or Blending), but rather to establish a simple, transparent, and reproducible baseline upon which future research can build.**
>
> We intentionally adopted a fixed ensemble mechanism across all experiments. This design choice allows us to perform a fair comparison of the learning strategies themselves (RL vs. GP vs. ML) by isolating the effect of the optimization paradigm from the impact of the ensemble architecture. Varying the architecture (e.g., comparing our RL-based weighting against a complex Stacking implementation) would introduce confounding factors, making it challenging to attribute performance gains to the learning principle itself.
>
> We acknowledge that methods like Stacking [1] or Weighted Voting [2] are relevant. Conceptually, our proposed strategy is similar to Weighted Voting: we weight detectors based on their reliability, but unlike standard global weighting, our weights are tailored to each time series via the weighting model (MLP or CNN) rather than fixed globally for the whole dataset.
>
> A key distinction also lies in the training workflow. Techniques like Stacking and Blending typically require complex data partitioning (nested cross-validation or out-of-fold predictions) to avoid leakage. **In contrast, our goal was to design a minimal baseline that does not depend on such complex partitioning strategies, facilitating easy reproducibility and adoption as a standard benchmark in the field.**
>
> References:
>
> [1] Wolpert, D. H. (1992). Stacked generalization. Neural Networks.
>
> [2] Khan, A. A., et al. (2024). A review of ensemble learning... Expert Systems with Applications.
>
> **Overall, thanks to this comment, we applied the following modification to the paper:**
> - We added a sentence in the introduction that refers to other ensembling approaches and justifies our experimental design : "Crucially, our goal is not to compare architectures(CNN vs LSTMs) or different ensembling methods
> (Stacking (Wolpert, 1992) vs Weighted voting (Khan et al., 2024)) or hyperparameter sensitivity,
> but to benchmark our strategies upon a simple, reproducible baseline for supervised ensembling
> grounded in existing research."

---

> ### Author Response · Authors · 2025-11-22
>
> > Sec 3.1: what are the statistical features?
>
> We thank the reviewer for this question, which allows us to detail the composition and motivation behind the use of catch22 features.
>
> Catch22 (CAnonical Time-series CHaracteristics) is not an arbitrary collection of features. It is a set of 22 descriptors selected from the HCTSA repository, which initially contained over 7,000 candidate features. These 22 features were identified through a rigorous data-driven filtering process aiming at:
> - Maximizing classification performance across the UCR archive.
> - Minimizing redundancy between features.
> - Ensuring computational efficiency.
>
> As a result, Catch22 provides a compact yet highly informative representation of temporal dynamics.
> Despite its small size, Catch22 covers a broad spectrum of signal properties. It captures both statistical and dynamical characteristics, including distributional properties (e.g., outliers and histograms), autocorrelation structures (e.g., the location of the first minimum), and entropy measures.
>
> Although Catch22 was initially optimized for time-series classification, it remains highly relevant for Anomaly Detection (AD), especially for supervised ensemble or model selection.  Indeed, the objective here is to find characteristics that we can match to specific detectors. Moreover, model selection for time-series anomaly detection has been studied as a time-series classification task [2]. Therefore, we believe that Catch22 is relevant to our case. Furthermore, Catch22 has been shown to be promising for time-series anomaly detection in a recent study [3].
>
> A critical advantage of Catch22 in our context is its compactness. Compared to exhaustive feature extraction libraries like TSFresh (up to ~1500 features), TSFEL, or Kats, Catch22 offers a significantly better trade-off between performance and dimensionality. This ensures good scalability for our ensemble methods (particularly RL and GP) without requiring extensive dimensionality reduction.
>
> References:
>
> [1] Lubba, C. H., et al. (2019). catch22: CAnonical Time-series CHaracteristics. Data Mining and Knowledge Discovery.
>
> [2] Sylligardos et al. (2023). "Choose Wisely: An Extensive Evaluation of Model Selection for Anomaly Detection in Time Series." (MSAD).
>
> [3] Tafazoli, S., Lu, Y., Wu, R., Srinivas, T. V. A., Dela Cruz, H., Mercer, R., & Keogh, E. (2024). C22MP: the marriage of catch22 and the matrix profile creates a fast, efficient and interpretable anomaly detector. Knowledge and Information Systems, 66(8), 4789-4823.
>
> **Overall, thanks to this comment, we applied the following modification to the paper**:
> - We have added a paragraph entitled "Features extraction and preprocessing." in Section A.5 describing in detail the computation of the feature and the handling of variable length time series for the raw representation

---

> ### Author Response · Authors · 2025-11-22
>
> > Sec 4.4: Any insights on why more activated detectors are better in out-of-distributions, while fewer activated detectors are better in in-distribution?
>
> We thank the reviewer for this insightful question regarding the varying effectiveness of ensemble size across different testing scenarios.
>
> **The observed difference stems from how the ensemble leverages detector diversity under distribution shifts.** In in-distribution settings, the test data closely matches the training distribution, thus the most relevant detectors (those that learned the specific training patterns) are highly reliable. In this context, incorporating additional, less capable detectors merely introduces noise, thereby reducing the overall signal-to-noise ratio. Consequently, a sparse selection of a few strong detectors yields the best precision, while "diluting" the best models with average ones degrades performance.
>
> In contrast, under out-of-distribution conditions, the reliability of any single detector drops significantly. **Thus, activating a broader set of detectors increases the diversity of anomaly scores**. This mitigates the risk of relying on a specific detector that might fail on the new data. **Aggregating many detectors provides a low-risk method, making the ensemble more robust to unexpected patterns.**
>
> **This asymmetry, where specialization favors in-distribution and diversity favors out-of-distribution, is consistent with prior observations in model selection literature.** As demonstrated in Sylligardos et al. (2023) [1], supervised model selection strategies (picking the "best" model) significantly outperform ensemble averaging in in-distribution settings. However, this advantage vanishes in out-of-distribution (leave-one-out) scenarios, where robust unsupervised ensembling prevails due to its lower variance.
>
> References:
>
> [1] Sylligardos, G., et al. (2023). Choose Wisely: An Extensive Evaluation of Model Selection for Anomaly Detection in Time Series. (Also related to MSAD framework).
>
> **Overall, thanks to this comment, we applied the following modification to the paper:**
>
> - We have enriched the last sentence of Section 4.4 : "The latter can be explained by the low sparsity of raw-based strategies (2 and 3 mainly) that mitigates the risk of relying on a specific detector that might fail on the new data."
>
>
> > Figure 5, x axis: Oralee -> Oracle
>
> Many thanks, we fixed it.

---

### Official Review · Reviewer_ZzxU · 2025-11-10

**Soundness:** 3
**Presentation:** 3
**Contribution:** 2
**Rating:** 4
**Confidence:** 4

**Summary:**

This paper addresses the challenge that no single anomaly detector consistently outperforms others across different datasets and anomaly types in time-series anomaly detection (TSAD). The authors explore three supervised ensembling strategies (classical supervised learning, reinforcement learning, and genetic algorithms) for learning detector weighting schemes. They propose a unified pipeline and evaluate these methods on the TSB-UAD benchmark, comparing them against individual detectors, unsupervised averaging ensembles, and model selection approaches. Through extensive experiments spanning 16 datasets and 12 detectors, the authors demonstrate that reinforcement learning with feature-based inputs achieves the best in-distribution performance, outperforming individual detectors and even the model-selection oracle on several datasets. Moreover, models trained on raw inputs exhibit strong generalization in the leave-one-dataset-out (out-of-distribution) setting.

**Strengths:**

* The paper provides a systematic and insightful comparison between model selection approaches, which are inherently limited by the best individual detector’s performance against the supervised ensembling methods that learn to combine multiple detectors.

* The experimental design ensures fairness: all strategies use the same architectures (MLP for feature inputs, CNN for raw data), consistent data splits (approximately 60/40 train–test), and uniform evaluation metrics (VUS-PR or its proxy). This setup isolates the impact of the learning strategy itself.

* Per–time–series analyses clearly demonstrate that supervised ensembles leverage the complementary strengths of individual detectors, rather than merely selecting one, thereby reinforcing the authors’ motivation.

* The scalability study, which covers training time, inference time, and the trade-off between the number of detectors and overall performance, is both practical and informative.

**Weaknesses:**

* The paper omits comparisons to widely used ensemble methods such as stacking, blending, or weighted voting. Without these, it is unclear whether the observed improvements stem from the concept of supervised ensembling itself or from the specific learning strategies explored (ML, RL, GP).

* The paper would benefit from situating itself more clearly within existing literature. Are there prior studies that applied supervised ensembling or stacking to time-series anomaly detection? If so, how does this work differ methodologically or empirically?

* The RL approach clips detector weights to  [0,1], while ML and GP use unconstrained (possibly negative) weights.  An ablation varying clipping or allowing signed RL weights would clarify the effect of this constraint.

* ML and GP models occasionally learn negative weights, yet the paper does not analyze their meaning or impact. Are such weights beneficial, or do they indicate overfitting or instability? Reporting constrained vs. unconstrained variants would strengthen the interpretation.

* The paper does not present random seeds, standard deviations, or confidence intervals for results. Given the stochastic nature of RL and GP, this omission makes it difficult to assess the robustness or reproducibility of performance gains.

* The computation of catch22 features is not described, nor is the handling of variable-length time series. Clarifying whether the series are truncated, padded, or resampled would improve methodological transparency.

**Questions:**

Please address the questions mentioned in the weaknesses.

---

> ### Author Response · Authors · 2025-11-22
>
> > The paper omits comparisons to widely used ensemble methods such as stacking, blending, or weighted voting. Without these, it is unclear whether the observed improvements stem from the concept of supervised ensembling itself or from the specific learning strategies explored (ML, RL, GP).
>
> We thank the reviewer for their valuable comments, which have helped us better highlight the specific scope and motivations of our work.
>
> **In this work, we introduce, to the best of our knowledge, the first baseline for supervised ensembling in time series anomaly detection.** Our primary goal is not to perform an exhaustive comparison of existing ensemble architectures (such as Stacking or Weighted Voting), but rather to establish a simple, transparent, and reproducible supervised ensembling baseline upon which future research can build.
>
> We agree that several ensemble methods exist, including Stacking [1, 2, 3], Blending [1], and Weighted Voting [2]. However, comparing these architectures is not the focus of this study. **Instead, we perform a fair comparison of the learning strategies themselves by isolating the effect of the learning paradigm (ML, RL, GP) from the effect of the ensemble architecture.**
>
> The supervised ensembling strategy we propose is intentionally elementary: for each time series, we compute detector weights based on its features (or first-window representation). Conceptually, this is similar to supervised weighted voting, with the key distinction that the weights are context-dependent (tailored to each specific time series) rather than global.
>
> **To address the comparison with the specific techniques mentioned,** the main difference lies in the data split. Stacking and Blending typically require a nested cross-validation or a dedicated validation set to train a meta-model on out-of-fold predictions to avoid leakage. **In contrast, our goal is to design a minimal baseline that does not depend on complex partitioning strategies, making it easier to implement and reproduce as a standard benchmark for future supervised ensembling methods in this domain.**
>
> We hope this clarifies the intention behind our methodological choices.
>
> **Overall, thanks to this comment, we applied the following modification to the paper:**
> - We added a sentence in the introduction that refers to other ensembling approaches and justifies our experimental design: "Crucially, our goal is not to compare [...] different ensembling methods
> (Stacking (Wolpert, 1992) vs Weighted voting (Khan et al., 2024)) [...] but to benchmark our strategies upon a simple, reproducible baseline for supervised ensembling grounded in existing research."
>
> References:
>
> [1] Wu, T., Zhang, W., Jiao, X., Guo, W., & Hamoud, Y. A. (2021). Evaluation of stacking and blending ensemble learning methods for estimating daily reference evapotranspiration. Computers and Electronics in Agriculture, 184, 106039.
>
> [2] Khan, A. A., Chaudhari, O., & Chandra, R. (2024). A review of ensemble learning and data augmentation models for class imbalanced problems: Combination, implementation and evaluation. Expert Systems with Applications, 244, 122778.
>
> [3] Wolpert, D. H. (1992). Stacked generalization. Neural Networks, 5(2), 241-259.

---

> ### Author Response · Authors · 2025-11-22
>
> >The paper would benefit from situating itself more clearly within existing literature. Are there prior studies that applied supervised ensembling or stacking to time-series anomaly detection? If so, how does this work differ methodologically or empirically?
>
> We thank the reviewer for their insightful comment. It allows us to clarify the specific gap in the literature that motivates our work.
>
> **To the best of our knowledge, supervised ensembling (and specifically stacking) has not been formally explored or benchmarked for Time-Series Anomaly Detection (TSAD).** Existing research on ensembling for anomaly detection predominantly emphasizes unsupervised aggregation strategies, with a heavy focus on simple score averaging.
>
> **Previous literature consistently highlights unsupervised averaging as a strong baseline.** Foundational work, such as Aggarwal (2015) [1], notes that in the absence of labels, simple averaging is often the most robust strategy for outlier detection. More recent comprehensive studies, such as Goswami et al. (2022) [3], confirm that simple averaging remains a strong baseline for unsupervised ensembling in time series. Moreover, in the context of recent automated benchmarks like AutoT-SAD [4], average ensembling (OE-Avg) is shown to perform competitively. While it is often outperformed by supervised model selection approaches (such as MSAD [2]) in fully supervised settings, it remains notably robust in unsupervised or out-of-distribution scenarios. Similarly,  Sylligardos et al. (2023) [2] reports that average ensembling is among the strongest baselines under leave-one-dataset-out evaluation.
>
> However, none of these studies investigates **supervised ensembling architectures** tailored to TSAD. This specific gap is what motivates our work.**Our goal is to provide the first systematic examination of a simple and practical supervised ensembling baseline.** By applying this baseline consistently across three established learning paradigms (ML, RL, GP), **we aim to provide a foundation for more sophisticated supervised ensembling or stacking methods for TSAD in future work.**
>
> **Overall, thanks to this comment, we applied the following modification to the paper:**
>
> - We added the following two references in Section 2.3 : "While unsupervised ensembling offers simplicity (Goswami et al.,
> 2022; Schmidl et al., 2024) and robustness [...]"
>
> References:
>
> [1] Aggarwal, C. C. (2015). Outlier Analysis. Springer. (Chapter on Ensembling).
>
> [2] Sylligardos et al. (2023). "Choose Wisely: An Extensive Evaluation of Model Selection for Anomaly Detection in Time Series." (MSAD).
>
> [3] Goswami, M., Challu, C., Callot, L., Minorics, L., & Kan, A. (2022). Unsupervised model selection for time-series anomaly detection. arXiv preprint arXiv:2210.01078.
>
> [4] Alpogon et al. (2024). "TSB-UAD: An End-to-End Benchmark Suite for Univariate Time-Series Anomaly Detection." (AutoT-SAD context).

---

> ### Author Response · Authors · 2025-11-22
>
> > The RL approach clips detector weights to $[0,1]$, while ML and GP use unconstrained (possibly negative) weights. An ablation varying clipping or allowing signed RL weights would clarify the effect of this constraint.
>
> We thank the reviewer for suggesting this ablation study to disentangle the effect of the weight constraints (clipping to $[0,1]$) from the RL strategy's intrinsic performance.
>
> **To address this concern, we performed additional experiments to align the constraints across all methods.** We re-evaluated the ML and GP baselines by applying the exact same constraints used for the RL agent (weights in $[-1, 1]$ followed by clipping to $[0, 1]$). In addition, we re-evaluated the RL agent with relaxed constraints in $[-1, 1]$, allowing negative weights.
>
> **The results of these experiments confirm our original conclusions regarding performance:**
>
> - Even when ML and GP baselines are subjected to the same $[0, 1]$ constraints as the RL agent, Strategy 2 (RL) remains the top-performing approach. **This confirms that the performance gain is driven by the RL policy's ability to adapt dynamically, rather than being an artifact of the constraint itself.**
>
> - **The hierarchy of performance across the strategies remains unchanged:** the choice of the learning strategy (RL vs. ML/GP) still prevails over the choice of data type (features vs. raw series) in the in-distribution case.
>
> **Regarding the "Relaxed RL" experiment, we confirmed that the $[0, 1]$ constraint is indeed necessary for convergence.** Allowing negative weights leads to significant instability in the optimization. The agent frequently converges to degenerate policies (e.g., assigning many weights to $-1$). This is likely due to the high variance introduced in the reward signal when subtracting detector outputs is permitted, which destabilizes the policy gradient updates.
>
> **These findings confirm that the $[0, 1]$ constraint is a necessary design choice for stable RL training in this context, and that the RL approach maintains its superiority even under fair comparison settings.**
>
> **Overall, thanks to this comment, we applied the following modification to the paper:**
>  - We have added the full details of this ablation study discussed above in the Appendix A.9 and A.10.
>
> - We have updated Figure 4 with aligned constraints across all methods
>
> > ML and GP models occasionally learn negative weights, yet the paper does not analyze their meaning or impact. Are such weights beneficial, or do they indicate overfitting or instability? Reporting constrained vs. unconstrained variants would strengthen the interpretation.
>
> We thank the reviewer for this insightful question regarding the role and impact of negative weights in the ML and GP approaches.
>
> Negative weights can serve a specific functional purpose rather than being artifacts. **They can act as a correction mechanism for the systematic biases of individual detectors.** Some detectors may be sensitive to specific noise patterns, leading to frequent False Positives. By assigning negative coefficients to these detectors, the ensemble effectively "subtracts" this correlated noise from the aggregate score.
>
> Regarding stability, we observe a fundamental difference between the paradigms: the optimization methods for ML and GP do not cause divergence when negative coefficients are present. In contrast, as noted in the Appendix, the RL agent struggles with negative actions. The inclusion of negative weights increases the variance of the reward signal (due to the subtraction of detector outputs), destabilizing the policy gradient search and leading to degenerate policies unless clipped.
>
> To verify whether negative weights indicate overfitting, we examined the generalization capability of the unconstrained models, particularly in out-of-distribution scenarios.
>
> If negative weights were fitting noise (overfitting) on the training set, performance should degrade significantly on unseen OOD data. We did not observe such signs of overfitting. Even in the OOD case, the unconstrained GP continues to outperform the robust unsupervised ensemble baseline. **This suggests that the negative weights capture transferable structural patterns (bias correction) rather than training-specific noise.**
>
> **Overall, thanks to this comment, we applied the following modification to the paper:**
> - We have updated Figure 4 with aligned constraints across all methods.
> - We also applied the constraint on the out-of-distribution scenario to discuss the overfitting aspect, we added the analysis in Appendix A.10. The analysis on the in-distribution scenario is in Appendix A.9.

---

> ### Author Response · Authors · 2025-11-22
>
> >The paper does not present random seeds, standard deviations, or confidence intervals for results. Given the stochastic nature of RL and GP, this omission makes it difficult to assess the robustness or reproducibility of performance gains.
>
> We thank the reviewer for highlighting the importance of statistical rigor. We agree that, given the stochastic nature of RL and GPs, reporting variability is essential to assess the robustness of our method properly.
>
> **In response to this feedback, we have significantly extended our evaluation of the supervised experiments by re-running the training process across five different random seeds.**
>
> We now report two additional analyses to quantify variability. We have reproduced Figure 4 to display the **VUS-PR accuracy (with error bars) across the five seeds per time series**, providing a visual representation of stability. In addition, **we added a table highlighting variability at the strategy scale with the variance (over the seeds) of the mean performance**.
>
> **These new statistically rigorous results confirm the robustness of our original claims.** Even when accounting for variance, we observe that:
>
> - **Supervised ensemble strategies consistently outperform both individual detectors and the robust unsupervised ensemble baseline.**
>
> - **Strategy 2 (RL) on features remains the top-performing approach**. It is the only supervised ensembling pipeline that significantly outperforms the best model selection methods reported in Sylligardos et al. (2023).
>
> The "hierarchy of influence" we described holds: the choice of strategy (RL vs. others) prevails over the choice of input data type (features vs. raw series) in the in-distribution case.
>
> **Overall, thanks to this comment, we applied the following modification to the paper:**
> - We have added table 2 in the main text assessing the variability of the strategies.
> - We have updated Figure 4 to incorporate the average accuracy over 5 seeds per strategy in the in-distribution scenario.
>
> >The computation of catch22 features is not described, nor is the handling of variable-length time series. Clarifying whether the series are truncated, padded, or resampled would improve methodological transparency.
>
> We thank the reviewer for this insightful remark regarding the computational details of our pipeline.
>
> The computation of the catch22 features is performed directly using the official Python library (pycatch22). This ensures a **standardized and reproducible implementation.** Moreover, as noted in Lubba et al. (2019) [1], **computational efficiency was one of the primary selection criteria** for the catch22 set (alongside predictive performance, low redundancy, and interpretability), making it highly suitable for our extensive benchmark.
>
> Regarding variable-length handling, we clarify that this issue pertains primarily to the raw signal representation, as feature-based representations naturally map any series to a fixed dimensionality.
>
> For the raw time series, we apply a segmentation strategy that avoids synthetic padding while preserving data integrity. When possible, time series are segmented into non-overlapping windows of equal length. When the total length of a series is not an exact multiple of the window size, we do not truncate the end of the time series. Instead, we allow the first two windows to partially overlap. This adjustment shifts the subsequent windows such that the final window ends precisely at the last time point of the series.
>
> **This procedure ensures that all windows used in subsequent analyses have identical length without introducing zero-padding artifacts.**
>
> **Overall, thanks to this comment, we applied the following modification to the paper:**
> - We have added a paragraph entitled "Features extraction and preprocessing." in Section A.5 describing in detail the computation of the feature and the handling of variable length time series for the raw representation.
>
> References:
>
> [1] Lubba, C. H., Sethi, S. S., Knaute, P., Schultz, S. R., Fulcher, B. D., & Jones, N. S. (2019). catch22: CAnonical Time-series CHaracteristics: Selected through highly comparative time-series analysis. Data Mining and Knowledge Discovery, 33(6), 1821-1852

---

### Author Response · Authors · 2025-11-30
**Summary of Revisions: Strengthening Experimental Robustness, Clarifying Scope, and New Visualizations**

**Dear Area Chair,**

We thank the reviewers for their constructive feedback. Based on their insightful comments, we have significantly revised the paper to improve its clarity, experimental robustness, and positioning. Below is a summary of the key modifications made to the manuscript.

**Strengthened Experimental Robustness and New Analyses**
* **New Variability Study (Main Text):** To demonstrate stability, **we added Table 2 in the main text** and **updated Figure 4**. Both exploits differently the performances across 5 seeds and confirm that Strategy 2 (RL) consistently outperforms unsupervised ensemble and the other supervised strategies.
* **New Visualizations (T-SNE, Appendix):** Addressing the request for deeper analysis, we added **Section A.12** featuring T-SNE visualizations. By labeling time series with their best detector, we analyze how the supervised ensemble groups time series, providing interpretability to our method.
* **Ablation Studies (Appendix):** We included full details of the ablation study comparing clipped vs. unconstrained detector weights in **Appendix A.9 and A.10**.



**Clarified Scope, Motivation and Metrics**
* **Scope Definition (Main Text):** **We refined the Introduction** to explicitly state that, **as supervised ensembling remains unexplored in this domain**, our goal is not to benchmark architectures (CNN vs. LSTM) or ensemble methods (Stacking vs. Voting), but to **conduct the first investigation** into whether **ML,RL or GP supervised baseline can surpass both the unsupervised ensembling ceiling and the supervised model selection approach using a reproducible framework grounded in existing research** (Sylligardos et al. 2023).


* **Value of OOD (Main Text):** In **Section 4.2**, we clarified that our Out-Of-Distribution (OOD) experiments serve as **a proxy for robustness in label-scarce scenarios, showing that supervision is transferable**.

* **Metric Robustness (Appendix):** **We updated Appendix A.5 to justify the choice of VUS-PR (Boniol et al., 2025)**, emphasizing its design for noisy, misaligned, or imprecise labels inherent to anomaly detection.

**Methodological Positioning and Distinctions**
* **Differentiation from MoE (Main Text):** **We added a crucial distinction in Section 3.1** explaining that unlike Mixture of Experts (MoE), our framework uses a diverse set of pre-trained algorithms.
* **Distinction Genetic programming/Reinforcement Learning (Main Text):** We highlighted the complementarity of Genetic Programming and Reinforcement Learning in **Section 3.4**, referencing the EARL framework (Moriarty et al., 1999).
* **Unsupervised Baselines (Main Text):** We added references (Goswami et al., 2022; Schmidl et al., 2024) in **Section 2.3** to better contextualize to the best of our knowledge **the only existing ensembling approach for time series anomaly detection :   unsupervised ensembling.**



**Enhanced Reproducibility and Details**
* **Feature Extraction (Main Text):** **We added a detailed paragraph in Section A.5** describing feature computation and the handling of variable-length time series.
* **Interpretation of Results (Main Text):** **We enriched Section 4.4** to explain how the low sparsity of raw-based strategies mitigates the risk of relying on a single failing detector and thus perform better in OOD settings.
* **Data Presentation (Main Text):** **We updated Table 1** to highlight MLP and CNN performances on model selection (Sylligardos et al. 2023) and **refined the caption of Figure 4** to rigorously define the error bars computation.

We believe these revisions comprehensively address the reviewers' concerns and emphasize the novelty of our work as **the first reproducible baseline for supervised ensembling applied to time series anomaly detection.**

Best regards,

The Authors

---

**References**

Boniol, P., Krishna, A. K., Bruel, M., Liu, Q., Huang, M., Palpanas, T., ... & Paparrizos, J. (2025). VUS: effective and efficient accuracy measures for time-series anomaly detection. The VLDB Journal, 34(3), 32.

Goswami, M., Challu, C., Callot, L., Minorics, L., & Kan, A. (2022). Unsupervised model selection for time-series anomaly detection. arXiv preprint arXiv:2210.01078.

Moriarty, D. E., Schultz, A. C., & Grefenstette, J. J. (1999). Evolutionary algorithms for reinforcement learning. Journal of Artificial Intelligence Research, 11, 241-276.

Schmidl, S., Naumann, F., & Papenbrock, T. (2024). AutoTSAD: Unsupervised Holistic Anomaly Detection for Time Series Data. Proceedings of the VLDB Endowment, 17(11), 2987-3002.

Sylligardos, E., Boniol, P., Paparrizos, J., Trahanias, P., & Palpanas, T. (2023). Choose wisely: An extensive evaluation of model selection for anomaly detection in time series. Proceedings of the VLDB Endowment, 16(11), 3418-3432.

---

### Meta-Review · Area_Chair_C2uA · 2026-01-06

**Summary:**

Reviewers acknowledged the practical motivation of supervised ensembling for time-series anomaly detection and appreciated the scale of the experimental study. However, several concerns persisted regarding conceptual novelty and methodological rigor. In particular, multiple reviewers questioned whether the proposed framework offers a substantive advance beyond existing adaptive weighting or MoE-style ensembling approaches, viewing the main contribution as an extensive benchmarking exercise rather than a new methodological insight. Additional concerns were raised about assumptions on label availability, the lack of direct comparisons to established supervised ensembling methods (e.g., stacking or weighted voting), and limited theoretical or empirical justification for why specific learning paradigms (e.g., RL) outperform others. While the rebuttal added analyses and clarifications, these issues were not fully resolved, and the contribution was ultimately judged to fall short of the ICLR acceptance bar.

**Reviewer Concerns:**

The rebuttal addressed several implementation and evaluation details, including clearer experimental protocols, additional ablation studies, variability analyses across random seeds, and improved explanations of feature extraction and detector weighting. These changes increased confidence in the reported empirical results and reproducibility.

However, core concerns remain outstanding. Reviewers continue to question the novelty and conceptual contribution of the framework relative to existing supervised ensembling, adaptive weighting, or MoE-style approaches, and the work still lacks direct comparisons to standard supervised ensemble methods such as stacking or weighted voting. In addition, assumptions about label availability and the limited theoretical justification for why particular learning strategies (e.g., RL) should generalize better than simpler alternatives were not fully resolved.

**Reviewer Scores:**

see above

---

### Decision · Program_Chairs · 2026-01-26

Reject